# HIV-1 capsid stability enables inositol phosphate-independent infection of target cells and promotes integration into genes

**Gregory A. Sowd** [ORCID]*, **Jiong Shi, Ashley Fulmer, Christopher Aiken***

Department of Pathology, Microbiology, and Immunology, Vanderbilt Institute for Infection, Immunology, and Inflammation, Vanderbilt University Medical Center, Nashville, Tennessee, United States of America

* Gregory.A.Sowd@vumc.org (GAS); Chris.Aiken@vumc.org (CA)

**Data Availability Statement:** Relevant data are included within the manuscript, the Supporting Information, and are deposited to Dryad (https://doi.org/10.5061/dryad.5mkkwh79k).

## Abstract

The mature HIV-1 capsid is stabilized by host and viral determinants. The capsid protein CA binds to the cellular metabolites inositol hexakisphosphate (IP6) and its precursor inositol (1, 3, 4, 5, 6) pentakisphosphate (IP5) to stabilize the mature capsid. In target cells, capsid destabilization by the antiviral compounds lenacapavir and PF74 reveals a HIV-1 infectivity defect due to IP5/IP6 (IP5/6) depletion. To test whether intrinsic HIV-1 capsid stability and/ or host factor binding determines HIV-1 insensitivity to IP5/6 depletion, a panel of CA mutants was assayed for infection of IP5/6-depleted T cells and wildtype cells. Four CA mutants with unstable capsids exhibited dependence on host IP5/6 for infection and reverse transcription (RTN). Adaptation of one such mutant, Q219A, by spread in culture resulted in Vpu truncation and a capsid three-fold interface mutation, T200I. T200I increased intrinsic capsid stability as determined by *in vitro* uncoating of purified cores and partially reversed the IP5/6-dependence in target cells for each of the four CA mutants. T200I further rescued the changes to lenacapavir sensitivity associated with the parental mutation. The premature dissolution of the capsid caused by the IP5/6-dependent mutations imparted a unique defect in integration targeting that was rescued by T200I. Collectively, these results demonstrate that T200I restored other capsid functions after RTN for the panel of mutants. Thus, the hyperstable T200I mutation stabilized the instability defects imparted by the parental IP5/6-dependent CA mutation. The contribution of Vpu truncation to mutant adaptation was linked to BST-2 antagonization, suggesting that cell-to-cell transfer promoted replication of the mutants. We conclude that interactions at the three-fold interface are adaptable, key mediators of capsid stability in target cells and are able to antagonize even severe capsid instability to promote infection.

## Author summary

The US FDA- and EU-approved compound lenacapavir targets the HIV-1 capsid protein (CA) to inhibit HIV-1 infection. Lenacapavir blocks infection at multiple stages of infection offering benefits over current antiretroviral therapies. Identification of other capsid

**Funding:** This work was supported by the Tennessee Center for AIDS Research Developmental Core Award (P30 AI110527) to G.A. S. and by National Institutes of Health (https:// www.nih.gov/) grants P50 AI150481, U54 AI170791, and R01 AI157843 to C.A. The funders had no role in study design, data collection and analysis, decision to publish, or preparation of the manuscript.

**Competing interests:** The authors have declared that no competing interests exist.

surfaces that can be targeted by next-generation inhibitors is of importance due to the high mutability of the virus. We sought to identify surfaces on CA that modulate capsid stability in the cell targeted for infection by HIV-1. We identified CA substitutions that render the virus dependent on host metabolites that promote capsid stability. Replication of the host factor-dependent virus resulted in alteration of the Vpu protein and a second site on CA that stabilized the capsid. The compensatory mutation suppressed defects associated with the parental mutation, including decreased infection, changes to lenacapavir antiviral efficiency, and redistribution of integration targeting, a characteristic that affects latency. Our study reveals that an interface between the hexameric CA units in the capsid adapts to conditions that destabilize the capsid and that this interface is a primary modulator of capsid stability in target cells. We propose targeting of our identified capsid surfaces with inhibitors to increase lenacapavir potency and efficacy in patients.

## Introduction

During maturation, the HIV-1 capsid protein (CA) assembles into approximately 186 hexamers and 12 pentamers to create a fullerene cone that protects the viral genomic RNA thereby promoting reverse transcription (RTN) [1]. Further, CA facilitates host/virus interactions that mediate cellular trafficking of the capsid and integration targeting [1]. Consequentially, CA mutations have varied effects on infection and can perturb multiple aspects of infection with a single mutation. HIV-1 mutants CA N57A, Q63A/Q67A, and G89V prevent nuclear entry by disrupting interactions with nucleoporins NUP153 and/or NUP358, leading to changes in integration site distribution (ISD) [2–7]. Conversely, disruption of CA binding to host factor Cleavage and Polyadenylation Specific Factor 6 (CSPF6) with the CA N74D mutation reduces integration into transcriptionally active regions of the genome in favor of integration into transcriptionally poor, lamina-associated domains (LADs) [8,9]. LADs are genomic regions that bind to the nuclear lamina and are transcriptionally repressed [9]. These transcriptionally active regions targeted by HIV-1 for integration lie in nuclear speckle-associated domains (SPADs), transcriptionally active genomic regions near clusters of splicing-related factors [10]. Integration into active transcription units decreases provirus entry into latency and, conversely, increases latent provirus reactivation [11–14].

HIV-1 capsid lattice stability can be affected both *in vitro* and in cells by altering intra- and/ or inter-CA hexamer interactions with mutations [15,16]. HIV-1 CA mutants with lower intrinsic stability *in vitro* are associated with RTN defects in cells (e.g., P38A, K170A, K203A, and Q219A [15]). Conversely, CA mutants with hyperstable capsids tend to undergo RTN but fail to enter into the nucleus (e.g., E45A and E128A/R132A) [15,17]. It is unclear how capsid stability in cells is affected by host factor binding, and how capsid stability affects processes other than RTN like integration targeting.

The host metabolites inositol hexakisphosphate (IP6) and inositol (1, 3, 4, 5, 6) pentakisphosphate (IP5) bind to R18 and K25 rings in the mature CA hexamer to promote capsid assembly and its stabilization during maturation [18–20]. The mature capsid lattice is highly unstable *in vitro*, however, addition of IP6 prevents uncoating, allowing RTN to occur in both permeabilized virions and purified HIV-1 cores [18,21,22]. These *in vitro* RTN products are capable of integrating into target DNA, indicating that they are full-length HIV-1 cDNA [21]. Thus, IP6 binding by the capsid appears to be critical for capsid function *in vitro*. IP5 and IP6 are generated by successive phosphorylation of lower inositol phosphate species and affect various cellular pathways [23]. The levels of IP5 and IP6 can be modulated, and IP5/IP6 (IP5/6) are depleted by knockout (KO) of inositol phosphate multikinase (IPMK) [24]. IP6 is

selectively depleted by KO of IP5 2- kinase (IPPK), the cellular enzyme that phosphorylates IP5 position 2 [24]. In contrast to *in vitro* results, depletion of cellular IP5 and/or IP6 via IPPK or IPMK KO has, at best, a modest effect on target cell infection by HIV-1, even in T cells, indicating that these metabolites are dispensable in the context of the target cell [24–27].

The US FDA- and EU-approved antiviral drug lenacapavir (LCV, initially GS-6207) binds to the viral capsid and prevents RTN, nuclear import, and integration [28]. LCV also disrupts integration targeting to transcriptionally active genes [28]. Similar to the less potent inhibitor PF74 that binds the same CA intra-hexamer interface as LCV, the step of infection that is inhibited by LCV depends on the dose, with low concentrations of LCV primarily inhibiting nuclear entry and high concentrations of LCV fracturing the capsid to prevent RTN [21,28–32]. We previously showed that a combination of IP5/6 depletion and capsid-destabilizing concentrations of LCV or PF74 increased CA inhibitor antiviral efficiency against HIV-1 in target cells, suggesting that factors beyond IP6 stabilize the capsid in cells [29]. Given the promise and potency of LCV combined with relative conservation of the CA protein compared to other HIV-1 proteins, identification of other capsid interfaces that can be targeted by small molecules is of high importance.

We hypothesized that other aspects of the HIV-1 capsid modulate stability upon IP5 and IP6 depletion in cells targeted for infection. We identified four CA mutants with intrinsically unstable capsids that are profoundly dependent on cellular IP5 and IP6 levels for infection and RTN. Repeated passage of one of the mutants resulted in acquisition of the CA substitution T200I and truncation of the Vpu protein. Combining CA T200I with each IP5/6-dependent CA mutant restored RTN nearly to that of CA T200I in IP5/6-depleted cells. Additionally, T200I rescued several other defects associated with the original IP5/6-dependent mutant capsids. Analysis of Vpu mutant viruses indicated that loss of BST-2 antagonism contributed to adaptation of the CA Q219A mutant virus for replication and that HIV-1 CA T200I/Q219A is replication competent independent of Vpu status. We conclude that adaptation of the three-fold axis of the capsid can suppress capsid stability defects imparted by dependence on IP6 for infection, thereby promoting target cell infection and growth in culture. Our results indicate that changes in the three-fold interface can rescue severely unstable capsids thereby promoting normal sensitivity to capsid inhibitors and ISD.

## Results

### A subset of unstable capsid mutants requires IP6 to undergo RTN in target cells

We previously reported that IP6 depletion by IPMK or IPPK KO increases the antiviral efficiency of HIV-1 inhibition upon addition of concentrations of PF74 or LCV that destabilize the HIV-1 capsid, suggesting that other properties of the HIV-1 capsid may determine HIV-1 dependence on IP6 in target cells [29]. To test this and to identify CA residues that affect sensitivity to IP6 depletion, a panel of 37 HIV-1-GFP (HIV-1$_{GFP}$) CA mutants was assayed for infection of WT or IPMK KO CEM cells previously transduced with a lentiviral vector expressing IPMK-Flag (IPMK KO$_{IPMK-Flag}$) or, as a control, the corresponding empty vector (WT$_{Vector}$ and IPMK KO$_{Vector}$, respectively) [24]. The panel of HIV-1$_{GFP}$ single- and multi-residue CA mutants altered 37 amino acids (roughly 16% of the CA protein) known to affect various capsid functions, including capsid stability, reverse transcription, nuclear entry, and interactions with specific capsid-binding host proteins [2,3,38–40,4,15,17,33–37].

We found that most of the CA mutants infected WT$_{Vector}$ and IPMK KO$_{Vector}$ cells to a similar extent (Fig 1A–1C). Four CA mutants (M144A, R162A, D166A, and K199A) exhibited a modest 2- to 3- fold increased sensitivity to IPMK KO (Fig 1A–1C). By contrast, four other CA mutants (P38A, K170A, K203A, and Q219A) were highly sensitive to IP5/6 depletion, each

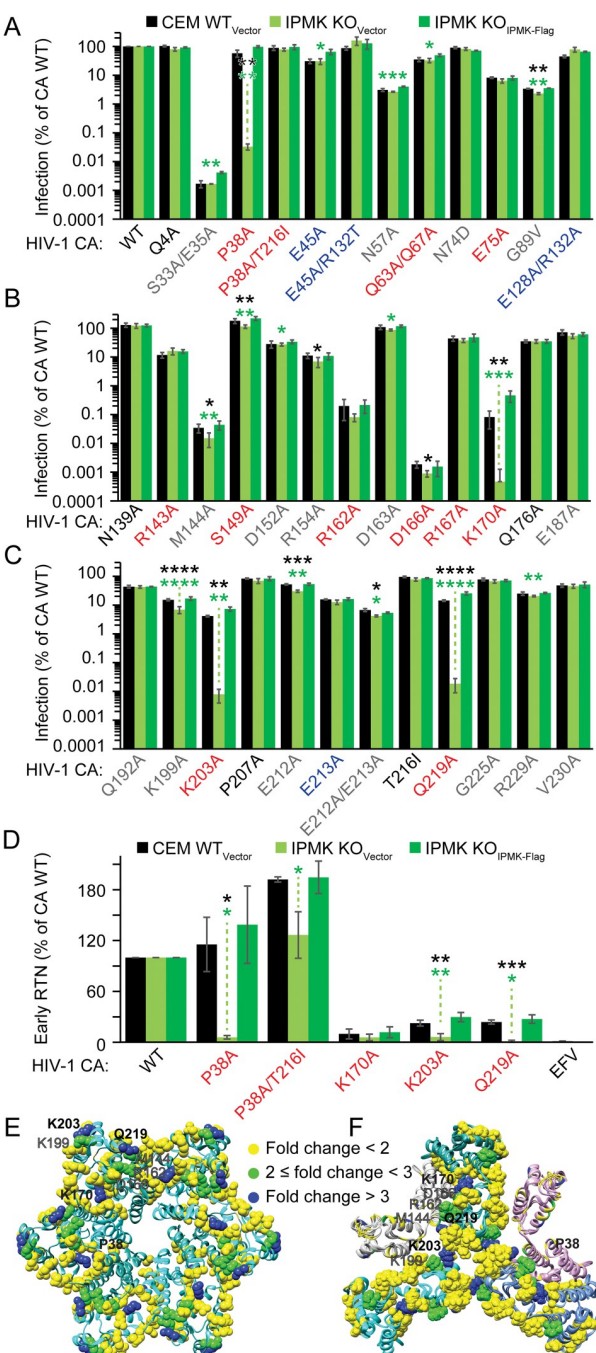

**Fig 1. Identification of IP6-dependent HIV-1 CA mutants.** (A, B, and C) Effect of CA mutations on HIV-1$_{GFP}$ infection of IPMK KO$_{Vector}$ CEM CD4$^+$ T target cells. (D) Quantification of HIV-1$_{GFP}$ early RTN products (first strand transfer (FST)) at 8 hours for the indicated IP6-dependent HIV-1 CA mutants. (E and F) Positions of altered residues that were tested for IP6 dependence mapped onto the HIV-1 CA hexamer (PDB 4XFX) (E) or the capsid three-fold interface (CA NMR structure (PDB 6WAP) aligned to the cryo-electron microscopic structure of the HIV-1 capsid (PDB 5MDG)) (F). Black asterisk(s) signify significance between IPMK KO$_{Vector}$ and WT$_{Vector}$ cells. A significant difference between IPMK KO$_{Vector}$ and IPMK KO$_{IPMK-Flag}$ cell lines is shown with a green asterisk(s). All data points represent the average of 3 or more independent experiments. p value levels: * $p < 0.05$, ** $p < 0.01$, *** $p < 0.001$, and **** $p < 0.0001$.

exhibiting a >100-fold decrease in infection in IPMK KO versus control cells (Fig 1A–1C). HIV-1 P38A containing its previously-identified suppressor mutation T216I [17] reversed the IP6 dependence of the HIV-1 P38A single mutant (Fig 1A). Although the IP6-dependent CA mutants P38A, K170A, K203A, and Q219A are known to reduce capsid stability *in vitro*, several other mutants tested are also unstable *in vitro* (e.g., Q63A/Q67A, E75A, R143A, S149A, and R167A), suggesting that capsid instability is necessary, but not sufficient for IP6 dependence (Fig 1A–1C) [15,36,41]. We conclude that infection by the HIV-1 CA P38A, K170A, K203A, and Q219A mutants depends on target cell IP6.

To identify the step in HIV-1 infection affected by IP6 depletion for these mutants, we quantified early RTN products produced by each IP6-dependent HIV-1 mutant in control vs. IPMK KO cells by quantitative PCR (qPCR). In CEM $WT_{Vector}$, RTN varied widely depending on the CA mutant tested with CA P38A and K170A being the least and most impaired, respectively (Fig 1D). For each IP6-dependent CA mutant and relative to either control cell line, IPMK KO decreased RTN to or near the background of the assay (Fig 1D). Addition of T216I to P38A restored RTN in IPMK KO cells to near that in WT cells, demonstrating that the defect is reversible. These results indicate that infection by these mutants fails due to a requirement for target cell IP5 and/or IP6 during RTN.

We next examined the location of the residues whose mutations engendered IP6 dependence to determine whether there are structural commonalities. The location of each residue mutated in our screen was mapped onto the reported structures of the CA hexamer and a model of the three-fold interface (Fig 1E and 1F). All CA mutations that conferred a >2-fold sensitivity to IP5/6 depletion were near intra- and inter-CA hexamer interfaces (Fig 1E and 1F). To examine the inter-CA interactions that may be affected by IP5/6-dependent CA mutations, an all-atoms model of the HIV-1 capsid (PDB 3J3Y [42]) was used to predict the inter-CA bonds at these residues (Figs 2A and 2B and S1). CA P38, R162, D166, M144 and K170 exclusively form intra-hexamer contacts (Figs 2A and 2B and S1C). Depending on the position of Q219 in the capsid, Q219 forms both intra- and inter-hexamer interactions (Figs 2A and 2B and S1C). By contrast, K199 and K203 interact with neighboring residues only at the three-fold inter-hexamer interface (Figs 2A and 2B and S1C). Thus, all IP6-sensitive and IP6-dependent HIV-1 CA mutations affected intra- and/or inter-hexamer CA interactions.

## IP5 can partially substitute for IP6 to promote infection by CA mutants

We previously showed that capsid destabilizing concentrations of CA inhibitors LCV and PF74 more efficiently inhibited HIV-1 in IPMK KO cells vs. IPPK KO cells [29]. To determine if the collection of IP6-dependent mutants is more sensitive to IPMK KO or IPPK KO, IPPK KO CEM target cells were infected with WT or IP6-dependent $HIV-1_{GFP}$. Relative to WT and IPPK $KO_{Flag-IPPK}$ cells, IP6-depleted cells were >8-fold less sensitive to infection by HIV-1 CA P38A, K170A, K203A, or Q219A (Fig 2C). Complementation of the IPPK KO cells line with IPPK rescued the infection of each IP6-dependent HIV-1 mutant to the extent that IPPK $KO_{Flag-IPPK}$ CEM target cells were 1.5 to 4-fold more permissive to each IP6-dependent CA mutant. By comparison to IPPK KO, IPMK KO decreased infection of IP6-dependent HIV-1 by >450-fold for each CA mutant relative to infection of either control cell line (Fig 2C). IPMK $KO_{IPMK-Flag}$ cells were more sensitive to infection by each HIV-1 mutant. Further, relative to depletion of IP6 alone, depletion of both IP5 and IP6 decreased HIV-1 CA P38A, K170A, K203A, or Q219A infection by an additional 40-, 10-, 64-, and 64-fold, respectively (Fig 2C). We conclude that target cell IP5 as well as IP6 can promote infection by HIV-1 CA mutants.

To confirm consistency of the phenotype, we next asked whether the IP6-dependent phenotype varies with the target cell line used for infection. IPPK KO, IPMK KO, and the respective

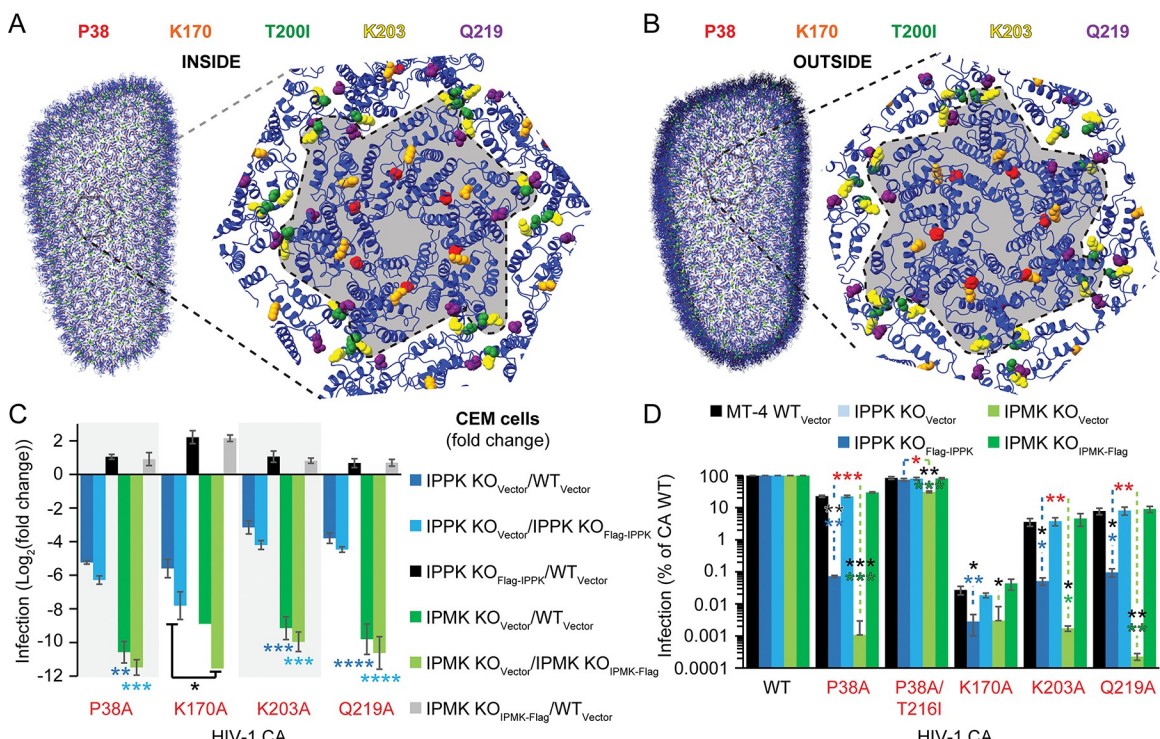

**Fig 2. IP5 can substitute for IP6 in target cell infection of IP6-dependent HIV-1.** (A and B) Relevant HIV-1 CA residue positions within an atomic resolution model of the HIV-1 capsid (PDB 3J3Y). The inner (A) and outer (B) capsid surfaces are depicted. (C) Comparison of HIV-$1_{GFP}$ infection of the indicated target cell lines as determined by flow cytometry. The fold change from the y-axis represents the fraction of HIV-1-infected cells for the indicated cell line divided by the percentage of infected cells of the denoted control cell line. The blue asterisk(s) indicate significance when comparing the of $Log_2$(fold change) for IPPK $KO_{Vector}$/WT$_{Vector}$ vs. IPMK $KO_{Vector}$/WT$_{Vector}$. The turquoise asterisk(s) denote significance between the $Log_2$(fold change) between the $KO_{Vector}$ cell lines when normalized relative to the respective complemented cell line. For K170A: The bars for IPMK$_{KO}$ use the average % of WT values to calculate the $Log_2$(fold change). When infecting IPMK $KO_{Vector}$ CEM cells, several HIV-1 CA K170A infections had no infected cells, thus prohibiting the calculation of $Log_2$(fold change) using the values from the independent experiments. For this mutant, the black asterisk indicates a significant difference when comparing infection of IPPK $KO_{Vector}$ and IPMK $KO_{Vector}$ cells. (D) HIV-$1_{GFP}$ CA WT or IP6-dependent CA mutant infection of the indicated MT-4 target cell lines as assessed by flow cytometry. A significant difference is denoted by asterisk(s) for the comparing infection of $KO_{Vector}$ cells to WT$_{Vector}$ cells (black) or complemented cells to $KO_{Vector}$ cells (blue or green for IPPK or IPMK KO, respectively). Red asterisks signify a significant difference when comparing infection of IPPK $KO_{Vector}$ cells to IPMK $KO_{Vector}$ cells. Each bar is the average of 3 or more independent experiments. Significance levels: * $p < 0.05$, ** $p < 0.01$, *** $p < 0.001$, and **** $p < 0.0001$.

control CD4$^+$ MT-4 cells lines were challenged with the panel of IP6-dependent HIV-1 CA mutants and assayed for single cycle infection. As in CEM cells and relative to respective controls, infection of all four CA mutants was reduced by IP5 and/or IP6 depletion in MT-4 cells (Fig 2D). Except for CA K170A, for which quantification of infection was limited by assay sensitivity (~1 infected cell per 300,000 cells), all IP6-dependent HIV-1 CA mutants were less infectious in IPMK KO cells compared to IPPK KO cells (Fig 2D). Thus, the IP6-dependent HIV-1 phenotype is independent of T cell line used. Henceforth, these HIV-1 CA mutant viruses will be referred to as IP5/6-dependent HIV-1.

## In vitro selection for HIV-1 CA Q219A replication selects for CA T200I and truncated Vpu

To identify other viral determinants that influence IP5/6-dependent HIV-1 fitness in T cells, we studied adaptation of the CA mutants by serial passage in T cell cultures. WT CEM cells

were inoculated with increasing amounts of full-length HIV-1$_{R9}$ CA Q219A. WT HIV-1$_{R9}$ replication peaked at 11 days when cells were inoculated with 1 or 5 ng p24. By contrast, HIV-1$_{R9}$ CA Q219A required an inoculum of 25 ng p24 to detect replication and release peaked at 36 days (Fig 3A). Sanger sequencing of the integrated HIV-1 provirus at 32 days post inoculation revealed that the genotype of this passage 1 (P1) virus was CA Q219A/Vpu W22Stop (Figs 3A and S2A). During passage 2 (P2), accumulation of RT activity in both of the duplicate cultures peaked at 11 days, retaining the Vpu mutation and CA Q219A, and evolving a second CA mutation, T200I, that became predominant during passage 3 (Figs 3A and S2B-S2E). A third, less abundant mutation, MA V128I, was detected in one of the cultures, but failed to accumulate (S2B–S2E Fig). All other HIV-1 loci remained unaffected by passaging. Thus, the HIV-1 CA Q219A virus evolved a truncated Vpu allele and the CA T200I mutation to adapt for replication in culture.

## CA T200I suppresses IP5/6-dependent RTN

To quantify the potential effects of T200I on infectivity of the IP5/6-dependent HIV-1 CA mutants, WT CEM target cells were infected with HIV-1$_{GFP}$ CA WT, HIV-1$_{GFP}$ CA T200I, or the panel of IP5/6-dependent HIV-1$_{GFP}$ with or without the CA T200I mutation and assayed for infection by flow cytometry for GFP positive cells. Addition of the CA T200I substitution decreased HIV-1$_{GFP}$ infection by 2.9-fold relative to HIV-1$_{GFP}$ CA WT (Fig 3B). Compared to the WT CA control, HIV-1$_{GFP}$ CA P38A, K170A, K203A, and Q219A infection was lowered by 1.8-, 294-, 9-, and 3.3-fold, respectively (Fig 3B). All IP5/6-dependent HIV-1 mutants containing CA T200I were rescued to the level of the HIV-1 CA T200I single mutant. Thus, a 73- and 3.7-fold increase in infection occurred for the K170A/T200I and T200I/K203A mutant viruses relative to the respective K170A and K203A single mutant virus (Fig 3B). We conclude that in WT cells the CA T200I rescues extremely unstable HIV-1 capsid mutants CA K170A and K203A.

We next examined if addition of the T200I substitution to each IP5/6-dependent HIV-1$_{GFP}$ CA mutant could affect infection of IP5/6-depleted cells. HIV-1 CA T200I infection was unaffected by IP5/6 depletion, while each of the four IP5/6-dependent HIV-1 CA mutant was sensitive to IP5/6 depletion in the single-round infection assay (Fig 3C). CA P38A, K170A, K203A, and Q219A mutations, combined with T200I, enhanced HIV-1 infection of IPMK KO$_{Vector}$ target cells by 230-, 4400-, 19- and 245-fold, respectively (Fig 3C). Thus, addition of the CA T200I substitution markedly rescues the impaired infectivity of IP5/6-dependent HIV-1 mutants in IP5/6-depleted cells.

To test if CA T200I resolves the RTN defect of IP5/6-dependent HIV-1 mutants in IPMK KO cells, we assayed RTN for a subset of the IP5/6-dependent HIV-1 CA mutants along with their respective T200I double mutants. Both early RTN products (first strand transfer (FST)) and late RTN products (full length minus strand (FLM)) were analyzed by qPCR. Relative to RTN by WT HIV-1, HIV-1 CA T200I generated increased quantities of early products; however, fewer late products were made in both cell lines, partially explaining its limited target cell infectivity (Figs 3D and S3B). As expected, in WT$_{Vector}$ cells, each IP5/6-dependent HIV-1 mutant produced fewer RTN products than WT virus, and IPMK KO further inhibited RTN of these mutants (Fig 3D). CA T200I substantially rescued RTN of HIV-1$_{GFP}$ K170A in WT$_{Vector}$ cells (Fig 3D). In WT$_{Vector}$ cells, both the CA T200I/K203A and T200I/Q219A HIV-1 mutants produced greater amounts of early RTN products than the corresponding single mutants. Yet for these mutants, late RTN was limited to approximately the level of the respective IP5/6-dependent single mutant (Fig 3D). Thus, CA T200I limits late RTN, even when combined with the IP5/6-dependent CA mutations.

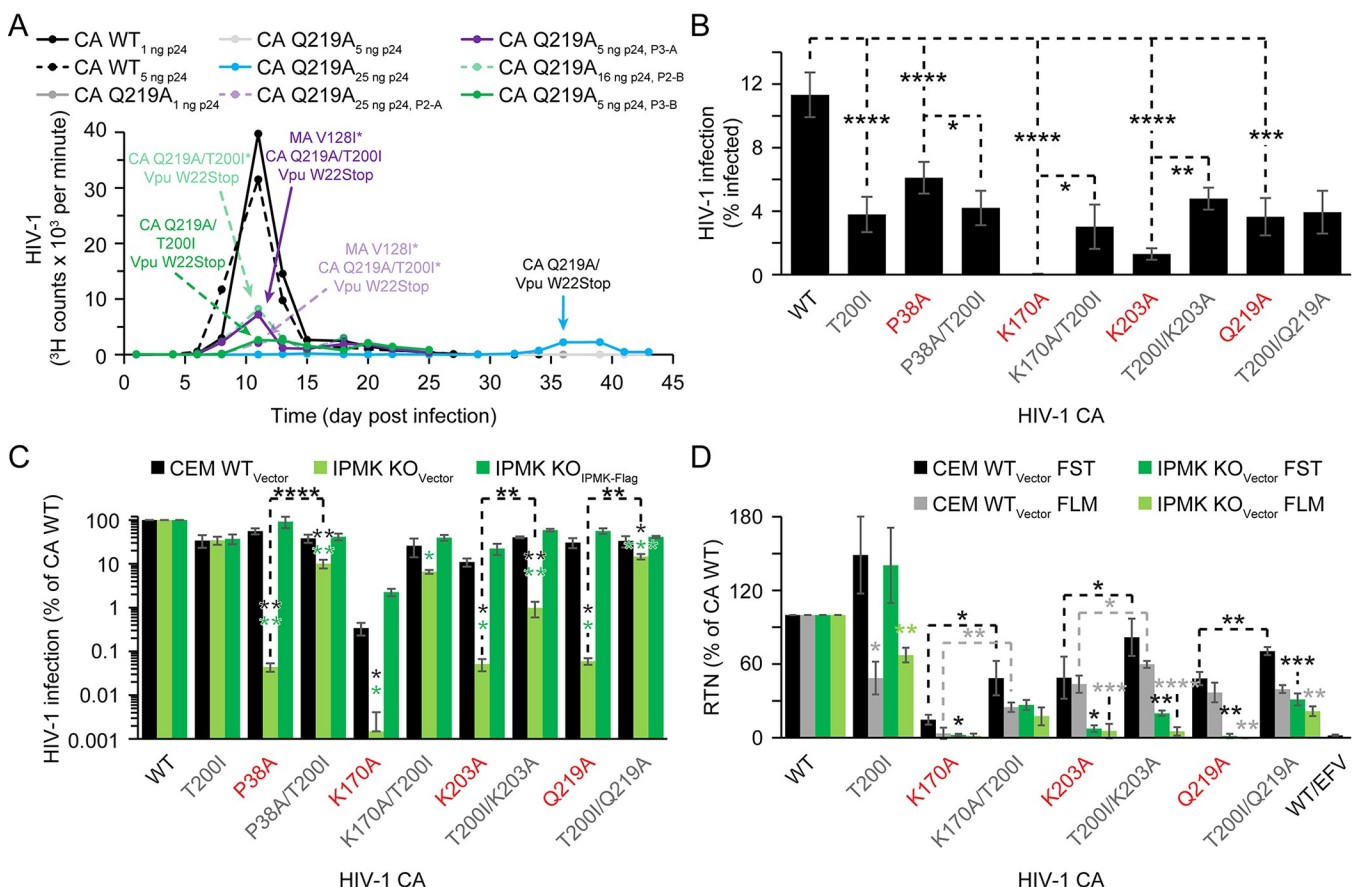

**Fig 3. CA T200I suppresses IP5/6 dependence by HIV-1 CA mutants.** (A) Passaging of HIV-1$_{R9}$ CA Q219A to select for adaptive mutations. Virus spread was monitored by exogenous RTN using the cell-free culture medium. Virus from the HIV-1 CA Q219A 25 ng p24 infection at ~ 36 days post infection was used to infect CEM cells to create passage 2 (P2) spread assays. P2-A and P2-B are independent P2 spread assays, in P2-B the virus was spin inoculated onto the cells. Passage 3-A (P3) and P3-B spread assays were inoculations of P2-A and P2-B, respectively, onto new CEM cells. Asterisks indicate that the indicated genotype was not the primary HIV-1 variant by Sanger sequencing. For genes with multiple genotypes (i.e. CA), mutations lacking an asterisk indicate the primary genotype. (B and C) Target cell infection of CEM WT cells (B) or the indicated cell lines (C) with the indicated HIV-1$_{GFP}$ reporter virus as determined by flow cytometry. (D) Relative amounts of early (FST) and late (full length minus (FLM)) RTN products. For asterisk(s) without comparison brackets, statistical significance within a given CA mutant is color coded and compares IPMK KO$_{Vector}$ to the corresponding (color-coded) control cell line. In (D), HIV-1 CA T200I statistics compare early and late products produced in the same cell line. All bars represent the average of 3 or more independent experiments. Significance levels: $p < 0.05$ *, $p < 0.01$ **, $p < 0.001$ ***, and $p < 0.0001$ ****.

By contrast to K170A, RTN by the K170A/T200I mutant was unaffected by IPMK KO, similar to WT HIV-1 (Figs 3D and S3B). In IPMK KO$_{Vector}$ cells, CA T200I provided no rescue to late RTN for the HIV-1 CA K203A/T200I mutant compared to its single mutant, implying that addition of T200I to the K203A single mutant might enhance a step in HIV-1 infection following RTN (Fig 3D). Conversely, relative to HIV-1 CA Q219A, addition of T200I nearly completely rescued Q219A RTN in IPMK KO$_{Vector}$ cells (Fig 3D). We conclude that T200I rescues RTN defects for the tested IP5/6 dependent mutants.

## CA T200I stabilizes the HIV-1 capsid

We next sought to obtain a better understanding of how T200I can rescue IP5/6-dependent HIV-1 mutants. As a first step to investigate the function of T200, we examined the spatial position of CA T200 in an all-atoms HIV-1 capsid model and mapped the nearby residues that

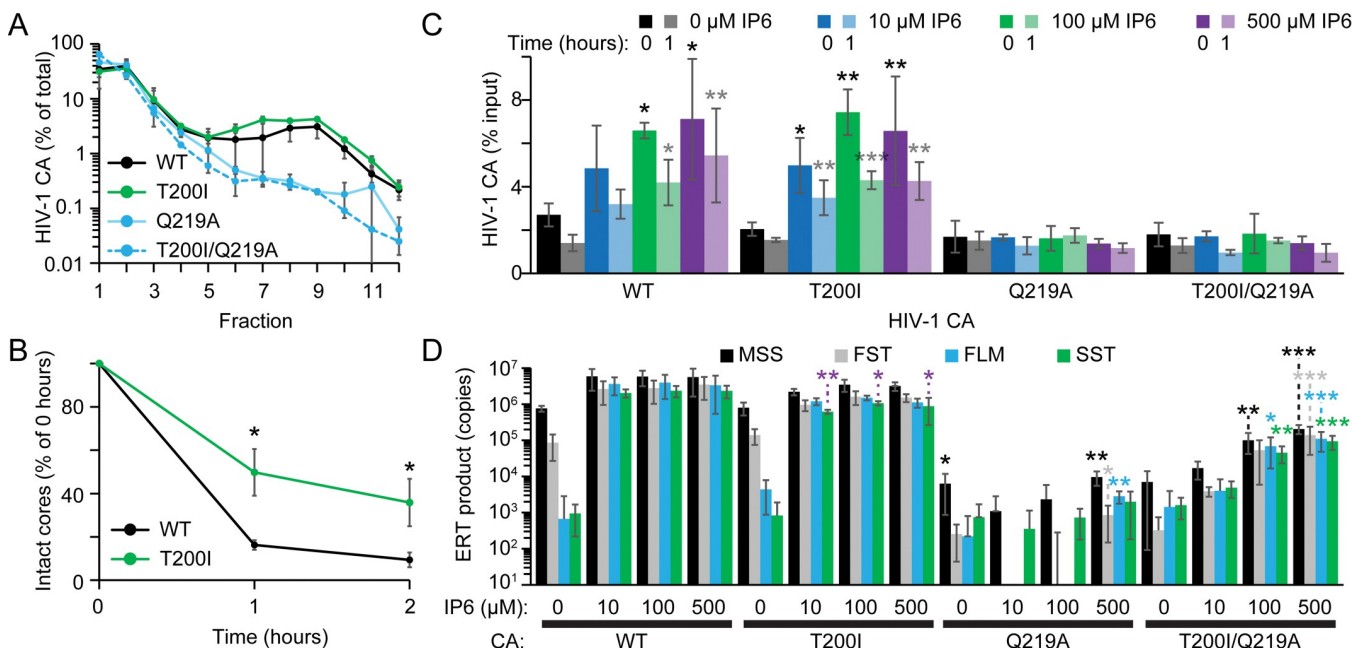

**Fig 4. CA T200I increases Q219A capsid stability.** (A) CA levels in sucrose gradient fractions of detergent-treated HIV-1 as assayed by ELISA. Fractions 7 to 9 normally contain intact HIV-1 cores. Data shows the average of 2 or more independent core purifications. (B) Uncoating of HIV-1 cores *in vitro* as quantified by p24 ELISA. (C and D) Assays of HIV-1 uncoating (C) and ERT reactions (D) in permeabilized virions. In (C), asterisk(s) denote significance when compared to the same virus timepoint lacking IP6. The purple asterisk(s) in (D) show significance when comparing ERT using WT and T200I cores. For Q219A and T200I/Q219A cores, asterisk(s) with colors matching the bars indicate significance of the results for the corresponding IP6 concentration vs. a reaction lacking IP6 for the same mutant core. In panels B through D, points show the average of values from 3 or more independent reactions. Significance level: $p < 0.05$ *, $p < 0.01$ **, $p < 0.001$ ***.

can form bonds [42]. T200 is located in the three-fold interface of the HIV-1 capsid (Fig 2A and 2B) and is predicted to form contacts with I201 and A204, as well Q219, residues determined to stabilize this interface (S1C Fig) [40,43]. Thus, we hypothesized that T200I acts by modulating the three-fold interface.

To address if T200I affected HIV-1 capsid stability in the context of Q219A, we attempted to purify cores from HIV-1 CA WT, T200I, Q219A, or T200I/Q219A virus using a linear sucrose gradient for use in uncoating assays [15,29,36,40,44]. WT and T200I cores exhibited similar amounts of CA protein in fractions 7 through 10, corresponding to intact cores (Fig 4A) [44]. By contrast, cores containing Q219A, even in the presence of T200I, could not be purified suggesting that the T200I/Q219A cores remain less stable than WT cores. The purified WT and T200I cores were analyzed for core stability using an uncoating assay which has been extensively used to assay for decreased or increased capsid stability [15,17,36,40]. Following dilution of purified WT HIV-1 cores into buffer, the core-associated CA protein was completely solubilized after a 1 hour incubation at 37°C (Fig 4B) [44]. Conversely, 3.1- and 3.8-fold higher quantities of CA remained associated with T200I mutant cores after 1 and 2 hours, respectively, relative to WT cores (Fig 4B). We conclude that the T200I mutation increases the intrinsic stability of the capsid, possibly explaining the observed rescue of infectivity and IP5/6 dependence exhibited by the CA mutant viruses.

We sought to determine whether the T200I mutation acts by stabilizing the Q219A mutant capsid. To this end, we incubated virions in buffer containing Triton X-100 and various concentrations of IP6. The reactions were then subjected to ultracentrifugation and the fraction of particle-associated CA protein was measured by quantifying CA protein in the pellet and supernatants by ELISA. In reactions lacking IP6, low amounts of core-associated CA were

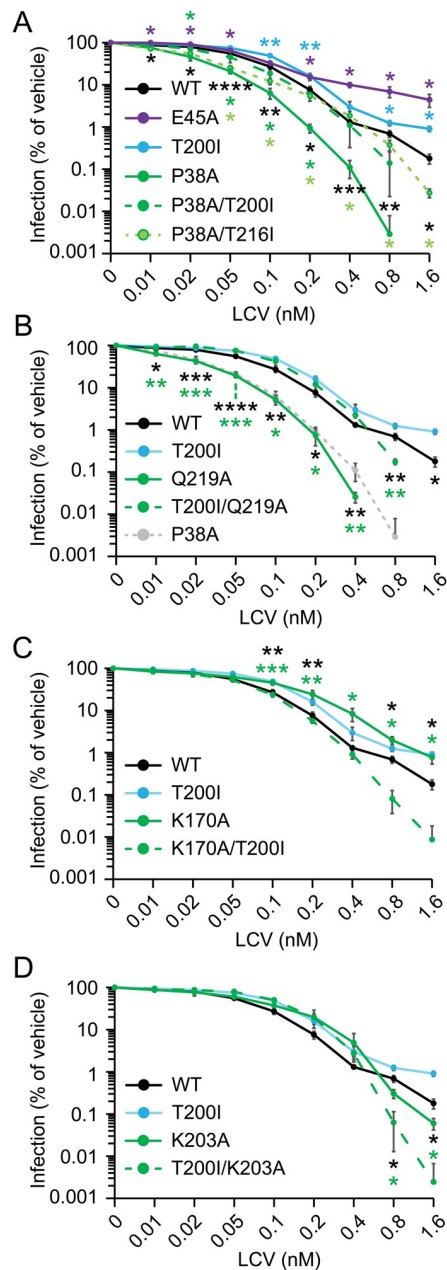

**Fig 5. CA mutation-associated effects on LCV antiviral potency are modulated by CA T200I.** (A–D)
Quantification of LCV inhibition of HIV-1$_{GFP}$ infection of CEM target cells as determined by flow cytometry. In (A),
significant differences between HIV-1 CA E45A or T200I and HIV-1 CA WT are denoted by purple and turquoise
asterisks, respectively. For all other dose response curve points, the asterisks are color coded to show which values are
being compared to HIV-1 CA WT. In (B-D), the HIV-1$_{GFP}$ CA WT and CA T200I curves are taken from (A). All data
points represent 3 independent experiments. Significance levels: $p < 0.05$ *, $p < 0.01$ **, $p < 0.001$ ***, and $p < 0.0001$
****.

retained by WT and T200I cores, yet incubation at 37°C lowered CA recovery to just above
background (~0.96% input) indicative of capsid solubilization (Fig 4C). Inclusion of IP6 in the
reactions increased the amount of pelletable CA, indicating that IP6 stabilized both WT and
T200I capsids (Fig 4C). Conversely, Q219A and Q219A/T200I mutant cores did not retain CA

protein, regardless of IP6 concentration (Fig 4C). These results indicate that T200I/Q219A cores remain less stable than WT cores *in vitro*.

We and others have previously reported that HIV-1 RTN *in vitro* is markedly dependent on capsid stabilization by IP6 [21,22,29]. Specifically, IP6 promotes synthesis of FLM and second strand transfer (SST) late RTN products (Fig 4D and [21,22,29]) in endogenous reverse transcription (ERT) reactions containing purified cores or permeabilized virions. Decreased DNA synthesis was observed in reactions containing IP6 concentrations <10 μM, which fail to fully stabilize the capsid [18,21,29,45]. Unstable capsid mutants are unable to undergo ERT independent of IP6 concentration used in the reaction [21]. Thus, the ERT assay can reveal differences in capsid stability in mutant HIV-1 particles [21]. To ascertain whether T200I stabilizes the Q219A mutant capsid *in vitro*, we assayed WT, T200I, Q219A, and T200I/Q219A virions in ERT reactions containing Triton X-100 a range of IP6 concentrations. In the absence of IP6, WT cores produced early minus strand strong stop (MSS) and FST products but few late products (Fig 4D). Addition of IP6 to ERT reactions containing WT cores stimulated late product synthesis by >3000-fold (Fig 4D).

At all IP6 concentrations tested, T200I produced similar amounts of MSS early product as WT but 2.5- to 3-fold fewer late SST late products (Fig 4D), consistent with the RTN defect observed in cells. DNA synthesis in Q219A particles was near background levels, and addition of 500 μM IP6, but not lower IP6 concentrations, increased SST synthesis by only 2.7-fold. By contrast, ERT in T200I/Q219A particles was progressively stimulated by IP6 in a concentration-dependent manner, with concentrations of 100 μM and 500 μM boosting SST synthesis by 22- and 58-fold, respectively, relative to the reaction lacking IP6 (Fig 4D). Considering that ERT only occurs using IP6 concentrations that stabilize the capsid [18,21], these results suggest that the T200I/Q219A capsid is more stable than the Q219A capsid but not as stable as the WT or T200I capsids.

## HIV-1 CA T200I affects HIV-1 sensitivity to LCV

Capsid stability was previously linked to HIV-1 sensitivity to the CA inhibitor PF74, as reflected by increased PF74 antiviral potency against the CA P38A and Q219A mutant viruses and the resistance exhibited by the hyperstable capsid mutant CA E45A, a mutation that has no effect on PF74 binding [30]. LCV functions similarly to PF74, but exhibits differences [28,29,31]. Thus, we tested how capsid stability in the context of our IP5/6-dependent HIV-1 mutants (+/-T200I) affects LCV inhibition of HIV-1 infection. CEM cells were infected with HIV-1$_{GFP}$ in the presence of various concentrations of LCV and infection was scored by flow cytometry. The hyperstable capsid mutants T200I and E45A exhibited resistance to LCV (Fig 5A and S1 Table). Each CA mutation led to substantial resistance to the compound at concentrations known to disrupt the capsid and inhibit RTN [29]. Conversely, the unstable mutants HIV-1 CA P38A and Q219A were hypersensitive to LCV at all tested concentrations (Figs 5A and 5B and S4A and S1 Table). Both the CA T200I and T216I suppressor mutations increased LCV inhibition of HIV-1 when combined with P38A and/or Q219A, albeit to different extents (Figs 5A and 5B and S4A and S1 Table). These results demonstrate that LCV potentiates with the CA P38A and CA Q219A substitutions to block HIV-1 infection. Because T200I stabilizes the HIV-1 capsid and reversed the LCV hypersensitivity of both the P38A and Q219A mutants, these results add further support to a mechanism of suppression by T200I that involves reversal of capsid instability.

By contrast to the P38A and Q219A mutants, the HIV-1 CA K170A mutant, which contains a highly unstable capsid [15,16], was less sensitive to LCV than HIV-1 CA WT. This effect was rescued by the K200I mutation (Figs 5C and S4B and S1 Table). Virus containing the CA

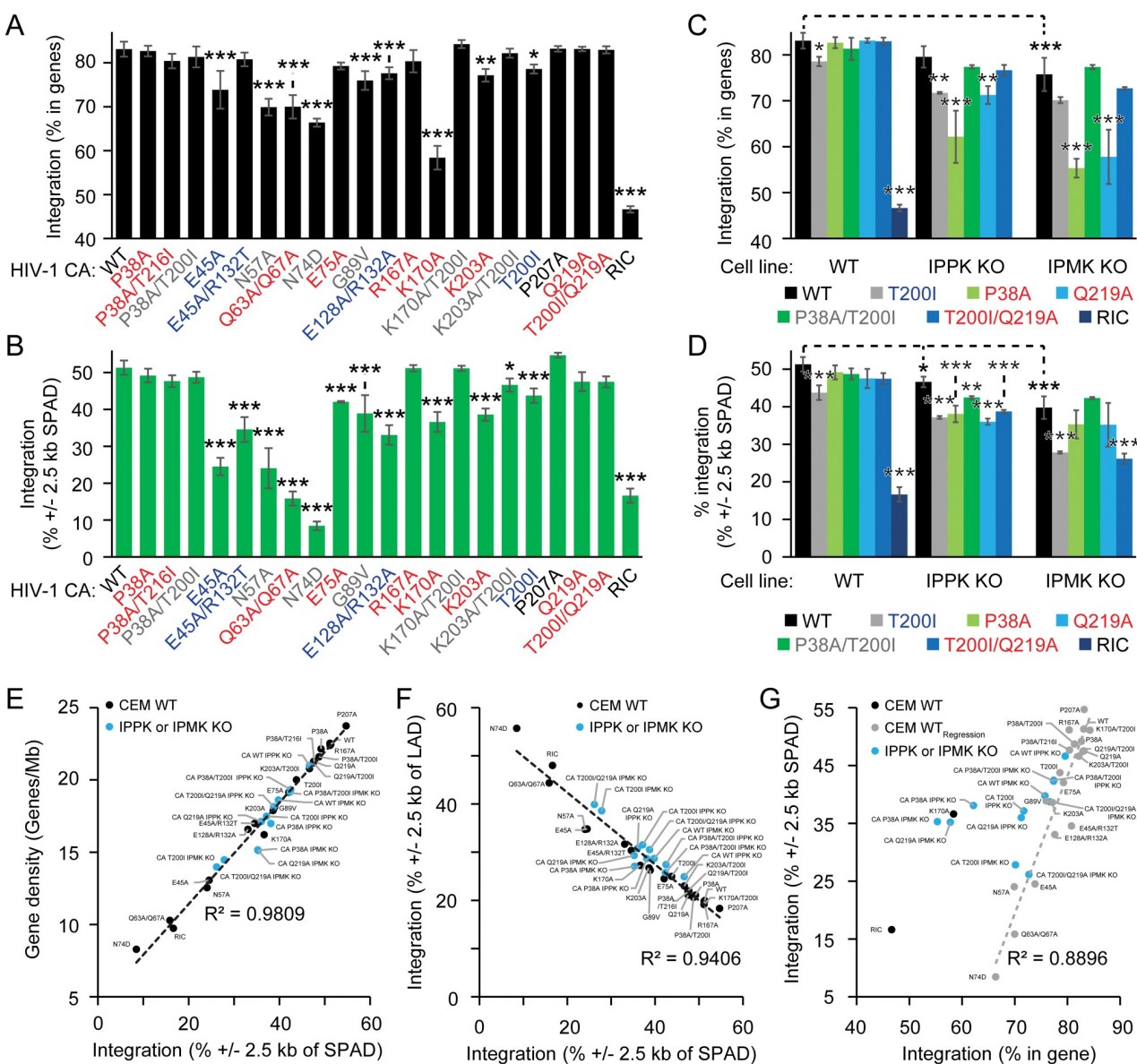

**Fig 6. Provirus integration site distribution is altered by capsid destabilization.** (A and B) HIV-1 integration targeting into the indicated annotations in CEM cells. In (A) and (B) asterisks show statistical significance relative to the HIV-1$_{GFP}$ CA WT control. (C and D) HIV-1$_{GFP}$ integration into genes (C) or near SPADs (D) in WT, IPPK KO, or IPMK KO CEM cells. For asterisk(s) lacking comparison brackets, each asterisk(s) compares to the HIV-1$_{GFP}$ CA WT control to the indicated mutant in same cell line. (E—G) Correlations between various genomic features used to determine the ISD. In (E) and (F), all points were used in the linear regression analysis. In (G), only the gray data points were used in the linear regression. The random integration control (RIC) is a computationally generated. All data are the average of 3 or more independent experiments. Significance levels: p < 0.05 *, p < 0.01 **, p < 0.001 ***, and p < 0.0001 ****.

K203A mutation was also less sensitive to LCV than the WT virus, except at concentrations >0.4 nM known to inhibit RTN (Fig 5D and S1 Table) [29]. Adding T200I to CA K203A created a virus that also exhibited decreased LCV sensitivity at concentrations below 0.4 nM, but the double mutant exhibited increased LCV sensitivity at concentrations > 0.4 nM (Figs 5D and S4B and S1 Table). We conclude that to a large extent CA T200I reverses the LCV sensitivity defects exhibited by the IP5/6-dependent HIV-1 mutants, regardless of whether the original mutant exhibits hypersensitivity or resistance.

## Hyperstable and severely unstable capsids retarget HIV-1 integration

We asked whether post-RTN aspects of infection are affected by CA mutations conferring IP5/IP6-dependence. Previous studies established that the magnitude of the ISD redistribution is indicative of the host/ virus process that is affected [2,6–8]. Defects in nuclear entry have less severe ISD redistributions than do disruptions that directly affect nuclear trafficking such as CA binding to CPSF6 [2,6–8]. Thus, the ISD was determined for the panel of IP5/6-dependent HIV-1 mutants (+/- T200I) and several other capsid stability mutants for comparison. Integration by WT and the nonimpaired P207A mutant [15,33] HIV-1 was nonrandom, with >80% and 50% of proviruses occurring in genes and SPADs, respectively (Fig 6A and 6B and S2 and S3 Tables). HIV-1 CA N74D, a CPSF6 binding mutant [39], decreased integration into genes by 17% and near SPADs by 43% (Fig 6A and 6B and S3 Table), a severe defect similar to previous studies [8,10]. Reducing nuclear entry with CA mutations N57A [3,46], Q63A/Q67A [3,35], G89V [3], E45A [17], or E128A/R132A [15,47,48] decreased integration into genes and near SPADs to varying extents consistent with the variety of defects leading to failed nuclear entry of these mutants (Fig 6A and 6B and S3 Table) [2]. The E45A suppressor mutation R132T [17] partially rescued integration targeting by HIV-1 CA E45A. HIV-1 CA T200I retargeted integration similarly to another hyperstable capsid mutant CA E128A/R132A and the CA mutant G89V, which exhibits impaired binding to Nup358 [2], a component of the nuclear pore complex (Fig 6A and 6B). Thus, we observed a spectrum of ISD patterns for the various CA mutants, with those exhibiting impaired nuclear entry showing less pronounced integration targeting changes than mutants with that affect CPSF6 binding [2,6–8].

Using HIV-1 CA mutants with a variety of capsid instabilities (K170A ≤ K203A < Q219A ≤ P38A < P38A/T216I < E75A < R167A < WT) [15,16,36], we next tested if an unstable capsid that endows a RTN defect is sufficient to affect ISD. HIV-1 CA E75A and R167A, which can be purified *in vitro* and have minor RTN defects [36], had little to no effect on ISD (Fig 6A and 6B and S3 Table). Similarly, HIV-1 capsid mutants CA P38A and Q219A, which have unstable capsids *in vitro* but have some stability in cells [15,16], exhibited WT ISDs (Fig 6A and 6B and S3 Table). However, HIV-1 CA K170A and K203A, capsids that are highly unstable *in vitro* and in cells [15,16,48], displayed integration targeting defects. HIV-1 CA K203A reduced targeting into genes and SPADs by 5.9% and 13%, respectively, compared to CA WT (Fig 6A and 6B and S3 Table). ISD was severely altered by HIV-1 CA K170A: integration into genes was decreased relative to CA N74D by 8% (Fig 6A). Nonetheless, HIV-1 pre-integration complexes (PICs) of the CA K170A mutant targeted SPADs at a much greater frequency than did the CA N74D mutant (Fig 6B). Addition of T200I to K170A or K203A rescued HIV-1 integration targeting to the level of CA WT (Fig 6A and 6B and S3 Table). These results hint that severe capsid instability leads to ISD defects and these defects are reversed by stabilization of the capsid by T200I.

## Complete loss of capsid stability imparts a unique defect in integration targeting

We next sought to determine if IP5/6 levels could affect the ISD of IP5/6-dependent HIV-1 mutants that had no initial integration site targeting defect to address if steps separate from RTN are affected by IP5- and/or IP6-depletion for these mutants. WT, IPPK KO, or IPMK KO CEM cells were infected with HIV-1 CA WT, CA P38A, or CA Q219A, and the ISD was determined. We observed that IP6 depletion decreased HIV-1 CA WT and CA T200I integration into genes and near SPADs, and combined IP5 and IP6 depletion led to a more pronounced ISD defect (Fig 6C and 6D and S4 Table). In IP5- and/or IP6-depleted cell lines, HIV-1 CA P38A and CA Q219A integration into genes was substantially decreased relative to HIV-1 CA

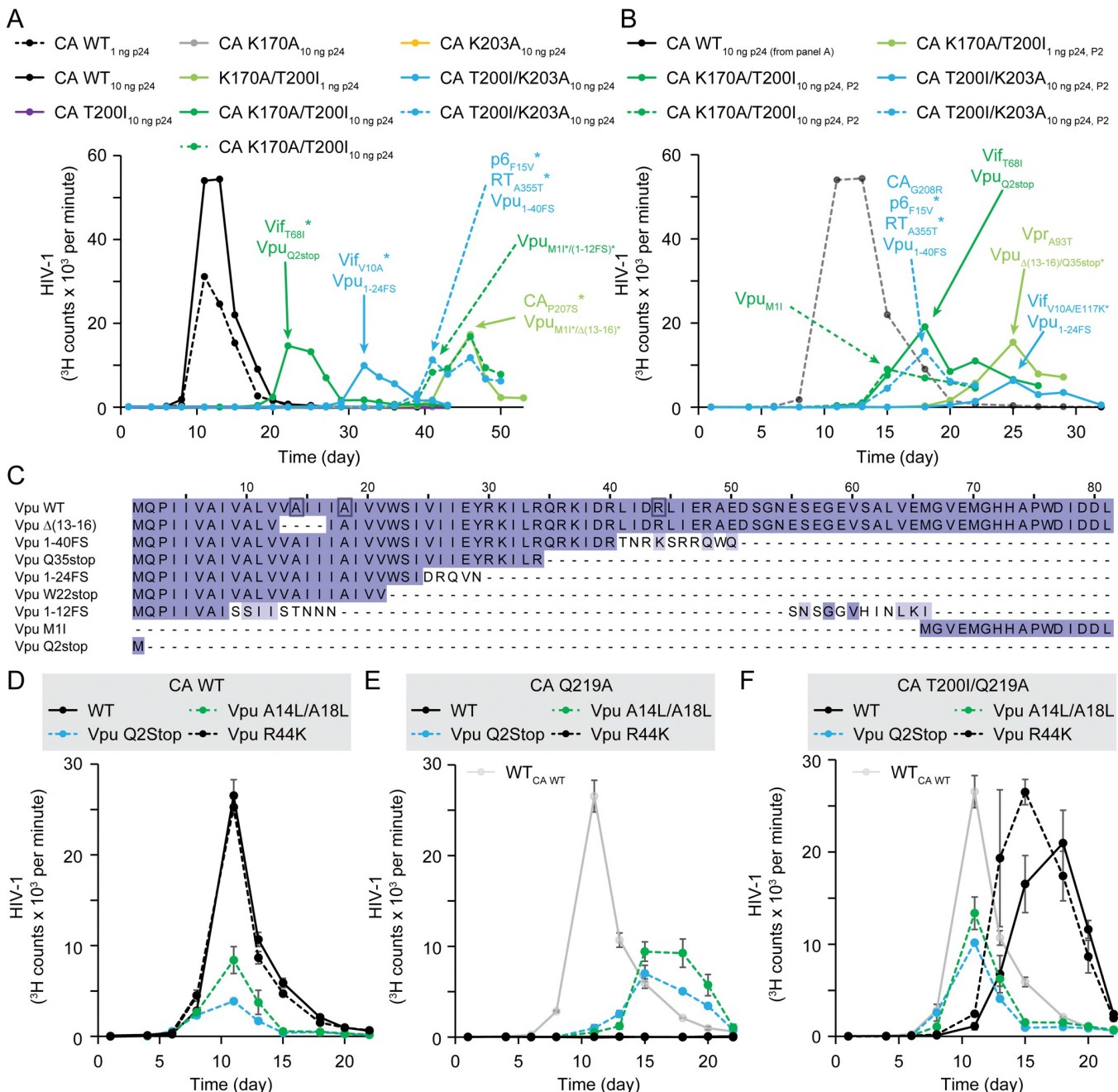

**Fig 7. Loss of Vpu antagonization of BST-2 promotes spread of replication-deficient HIV-1 CA mutants.** (A and B) HIV-1$_{R9}$ replication in CEM cells as monitored by RT activity in the cell culture media. Panel (A) shows P1 replication, whereas panel (B) shows P2 replication. In (B), point and line colors match panel (A) and indicate which P1 virus was used to initiate infection during P2. (C) Alignment of WT and truncated Vpu proteins identified during HIV-1$_{R9}$ replication experiments in Figs 3A and 6A and 6B. Indigo highlighting represents residue conservation with the WT Vpu sequence. (D–F) Effect of Vpu mutation on HIV-1$_{R9}$ CA WT (D), CA Q219A (E), or CA T200I/Q219A (F) replication. The average of 2 independent infections is shown in panels D to F.

WT (Fig 6C and S4 Table). Addition of CA T200I to either IP5/6 dependent HIV-1 CA mutant restored integration into genes in both of the IPMK and IPPK KO cell lines to the level of WT virus (Fig 6C and S4 Table). Conversely, relative to WT virus, integration near SPADs was disrupted by IP5 and/or IP6 depletion for each IP5/6-dependent CA mutant. In each IPPK KO

and IPMK KO cell line, addition of CA T200I to HIV-1 CA P38A and CA Q219A restored integration near SPADs at or near the level of CA WT or CA T200I, respectively (Fig 6D and S4 Table). These results indicate that capsid disruption caused by the combination of these CA mutations and IP5/6 depletion results in a defect in integration targeting and that the stabilizing effects of T200I can counteract this effect.

We next compared the HIV-1 ISD defects observed in IP5/6-depleted cells to those identified in WT cells for the other CA mutants for various annotations. Integration near SPADs was positively and negatively linearly correlated with gene density ($\rho = 0.99$) and integration near LADs ($\rho = -0.96$), respectively, indicating that these annotations measure the same aspect of integration targeting: CPSF6-dependent integration into transcriptionally active genes (Fig 6E and 6F) [8–10]. Excluding the severely unstable CA K170A mutant, integration of all HIV-1 CA mutants tested into the genome of WT T cells followed a linear relationship with respect to integration into genes and near SPADs (Fig 6G, $\rho = 0.85$). However, HIV-1 CA P38A and CA Q219A integration into IP5- and/or IP6-depleted cells (and HIV-1 K170A integration) did not follow this correlation for these two annotations, restricting proviruses to ~35% integrations near SPADs and lowering integration into genes near or below the level of CPSF6 binding-defective CA N74D mutant (Fig 6G and S4 Table). Thus, we conclude that, upon IP5/6-depletion, IP5/6-dependent HIV-1 mutants have a novel ISD akin to K170A integration targeting caused by a complete destabilization of the capsid. These results indicate that aspects of infection downstream of RTN are affected upon IP5/6-depletion for these mutants.

## Loss of BST-2 antagonization by Vpu promotes growth of IP5/6 dependent HIV-1 CA mutants

Since CA T200I improved the single round fitness of K203A and K170A, we next addressed if HIV-1 CA K170A/T200I and T200I/K203A were able to replicate in culture. CEM cells were inoculated with 1 or 10 ng p24 of HIV-1$_{R9}$ CA WT, T200I, K203A, K170A, K170A/T200I, or T200I/K203A. HIV-1 replication was monitored by testing for virus release into the culture supernatant. HIV-1 CA WT replicated well at both inocula, and replication peaked at 11 days. Regardless of inocula, each single CA HIV-1$_{R9}$ mutant virus was defective for replication (Fig 7A). At the 1 ng p24 dose of HIV-1, three out of four HIV-1 CA double mutants failed to grow. The single HIV-1$_{R9}$ CA K170A/T200I that did grow took > 34 days to be detected in the culture media (Fig 7A). Increasing the inoculum to 10 ng p24 facilitated the growth of all HIV-1 CA K170A/T200I and CA T200I/K203A viruses (Fig 7A). Peak virus replication varied greatly between independent infections, suggesting that the viruses required further evolution to replicate.

To validate adaptation of the mutants, the P1 viruses were re-inoculated onto WT CEM cells, resulting in the emergence of HIV-1$_{R9}$ K170A/T200I or T200I/K203A variants (Fig 7B). Sequence analysis of one of the T200I/K203A P2 virus cultures identified a third CA substitution, G208R, that was the dominant genotype (Fig 7B, turquoise dashed line). CA G208R was previously observed to facilitate replication of HIV-1 CA H62A virus [49], suggesting that the T200I/K203A virus required further adaptation to replicate. All P1 and P2 viruses also had truncating mutations in Vpu (Fig 7A–7C). Eight Vpu truncation variants were identified from the various CA Q219A, K170A/T200I, and T200I/K203A proviruses (Fig 7C). Vpu increases HIV-1 release by antagonizing the host protein BST-2 (Tetherin) [50], modulates cell surface CD4 levels [51], and inhibits innate immune signaling [52]. Based on the regions truncated, each Vpu variant lacks regions known to be required for Vpu function [53,54]. Thus, the T200I substitution in CA, combined with the loss of Vpu function, appears to enable HIV-1 K203A and K170A replication in culture.

To identify the Vpu function that inhibits IP5/6-dependent HIV-1 mutant replication, we inserted altered Vpu genes into HIV-1$_{R9}$ CA WT, Q219A, and T200I/Q219A molecular clones. The substitutions inhibit the ability of Vpu to sequester BST-2 (A14L/A18L), to suppress innate immune signaling (R44K), or to be translated (Vpu Q2Stop) [52,55]. Analysis of the growth kinetics of the resulting mutant viruses in WT CEM cell cultures showed that HIV-1$_{R9}$ CA WT replication kinetics were unaffected by the Vpu mutations, though the mutants that fail to antagonize BST-2 accumulated to lower maximum levels (Fig 7D). By contrast, HIV-1$_{R9}$ CA Q219A virus encoding WT or R44K mutant Vpu genes failed to replicate. However, HIV-1$_{R9}$ CA Q219A viruses encoding the Q2Stop or A14L/A18L mutant Vpu proteins emerged in the cultures, suggesting that loss of BST-2 antagonization by Vpu enabled replication of the CA Q219A mutant virus (Fig 7E). All HIV-1$_{R9}$ CA T200I/Q219A mutant viruses tested replicated in the cultures, demonstrating that HIV-1$_{R9}$ T200I/Q219A is replication competent (Fig 7F). As with CA Q219A mutants, loss of Vpu-dependent BST-2 antagonism was associated with accelerated replication of the CA mutant viruses (Fig 7F). Thus, inclusion of the Vpu A14L/A18L mutation promoted or accelerated growth of CA Q219A-containing mutants. These results suggest that cellular BST-2 promotes the replication of HIV-1 mutants encoding the CA Q219A substitution.

## Discussion

Unlike *in vitro* assays in which IP6 appears to be critical for capsid stabilization, we and others previously demonstrated that depletion of the metabolite and its precursor were to a large extent dispensable for target cell infection [24,25,27,29]. Our data reinforce this point for WT virus, as neither infection nor RTN were affected by IP5/6 depletion (Figs 1 and S3). Of the 11 unstable capsid mutants we tested for IP5/6-dependence, 4 mutants that disrupt intra- and/or inter-hexamer interactions were reliant on IP5/6 in target cells, arguing that IP5/6-dependence is not due to the disruption of a unique interface (Figs 1, 2 and S1). Further, the capsids of HIV-1 CA P38A, K170A, K203A, and Q219A have differing intrinsic stabilities and RTN defects [15,16]. Thus, capsid instability is necessary but not sufficient to endow IP5/6-dependence on HIV-1.

If a distinct inter/intra-hexamer CA interface does not explain the IP5/6-dependence of the subset of CA mutants, then why do these mutants require the metabolites? We suggest that upon entry into target cells the IP5/6 bound to the CA hexamer inside the virions [25,56] is exchanged with target cell IP6 (Fig 8A and 8B, III). Consistently, despite initially containing IP6 originating from the producer cell [25], HIV-1 cores are unstable when little or no IP6 is present in the solution (Fig 4B) [25,29], suggesting that at insufficient IP6 concentrations hexamer-bound IP6 is lost resulting in core disassembly. Addition of as little as 1 µM IP6 prolongs the capsid half-life to >1 hour and higher IP6 concentrations further increase the capsid half-life [18,29], hinting that core-associated IP6 from in the viral particle is exchanged with IP6 in the target cell (Fig 8A and 8B, III).

By contrast to the *in vitro* results, in target cells the WT capsid remains stable independent of the amount of IP6 indicating that other forces negate the need for IP6 (Fig 8B, III). Previously, we demonstrated that for HIV-1 CA WT virus IP5- and/or IP6-depletion sensitizes the virus to CA inhibitors which supports this notion [29]. Further, for the IP5/6-dependent HIV-1 CA mutants, when IP6 dissociates, our data indicate that the combined loss of IP5/6 from the hexamer along with disruption of intra- and/or inter-hexamer interaction(s) is sufficient to cause the capsid to dissociate causing RTN failure (Fig 8B, VII). A single suppressor mutation at the three-fold interface, T200I, compensates for the loss of the intra- and/or inter-hexamer CA interaction(s) by stabilizing the capsid (Figs 3D, 5 and 6), rendering infection independent

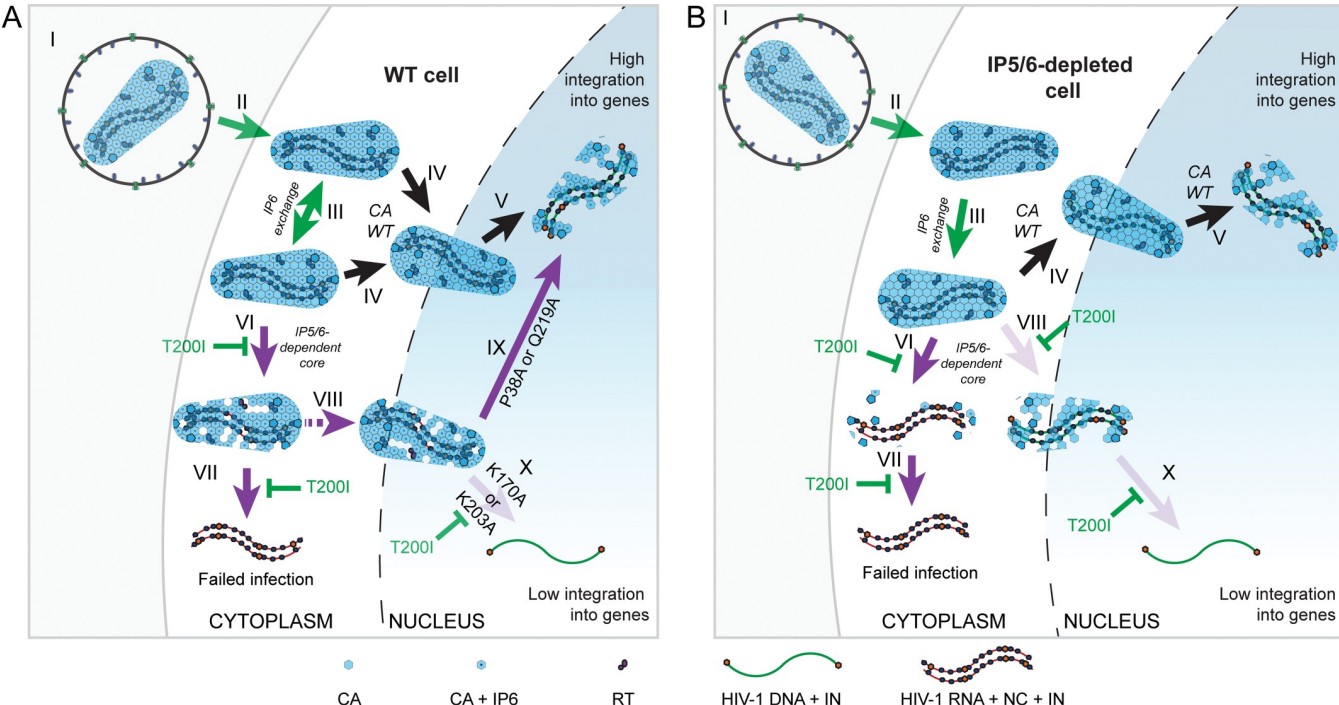

**Fig 8. Model of the role of IP6 in HIV-1 target cell infection.** (A and B) HIV-1 infection of WT (A) and IPMK/IPPK KO (B) cells. Panels (A and B): (I and II) The HIV-1 virion (I) enters the cell (II). (III) A fraction of capsid-bound IP6 dissociates and target cell IP6 reassociates with the CA hexamer (IP6 denoted by circle in the middle of the CA). (IV and V) For WT cores, regardless of the cellular IP5/6 concentration, the core enters the nucleus (IV), undergoes reverse transcription, and integrates into transcriptionally active genes (V). (A, VI to VII): For IP5/6-dependent HIV-1 mutants in WT cells, the exchange of IP6 can result in loss of CA monomers/hexamers (VI), leading to premature uncoating and impaired RTN (VII). (B, VI to VII): In IP5/6-depleted cells, premature uncoating is exacerbated, resulting in more extensive (or accelerated) capsid uncoating. (A and B, VIII) Independent of cell line, cores that survive premature uncoating enter the nucleus. (A, IX) In WT cells, more stable IP5/6-dependent mutant cores (CA P38A and Q219A) integrate into transcriptionally active genes. (A and B, X) Upon complete capsid destabilization, the PIC is unable to integrate within genes possibly owing to a dearth of CA. T200I along with a turnstile symbol indicates a step that is reversed by the mutation for the IP5/6-dependent HIV-1, including premature uncoating (VI), IP5/6-dependence, and decreased integration into genes (X). Arrow colors: Green–event occurs for HIV-1 CA WT and IP5/6-dependent HIV-1 CA mutants. Black–process shown happens for IP5/6-dependent virus containing T200I and HIV-1 CA WT. Purple–the event is only for IP5/6-dependent virus. Purple dashed line–efficiency of process depends on IP5/6-dependent CA mutant used. Opaque and faint purple arrows indicate more and less frequent events, respectively.

of (or minimally dependent on) target cell IP5/6 (Figs 3–6). CA T200I cores additionally have increased stability compared to WT in the absence of IP6 (Fig 4B). Thus, for the T200I double mutant capsids, IP6 may dissociate and reassociate independent of the amount of IP5/6 in the cell without loss of capsid stability in the cell (Fig 8, III and IV). Therefore, we conclude that the capsid three-fold interface of HIV-1 makes the virus generally independent of (or minimally dependent on) IP5/6 in target cells and is a modulator of capsid stability. All of this considered, additional host factors can stabilize the capsid *in vitro* and likely in cells (e.g. SEC24C [57] and CYPA [58,59]), likely negating the requirement for IP6 for HIV-1 CA WT as has been suggested elsewhere [29]. Thus, we suggest that the combination of intrinsic capsid stability with CA binding to host factors make the capsid sufficiently stable in cells to circumvent the *in vitro* requirement for IP6.

## The three-fold interface adapts to provide capsid stability

The dependence on IP5 and IP6 by the set of unstable HIV-1 capsid mutants underscores the delicate balance that the virus must maintain to allow RTN to occur, while preventing premature uncoating. The emergence of CA suppressor mutations T200I, G208R, and T216I at the three-fold interface during HIV-1 CA Q219A, H62A, T200I/K203A, or P38A replication in

culture reinforces this point (Figs 3A, 7A and 7B, and [17,49]). Of these suppressor mutations, only T216 forms both intra- and inter-hexamer contacts and is mainly predicted to form intra-hexamer contacts (S1C Fig). Conversely, T200 forms contacts with I201 and A204, residues reported to stabilize the three-fold interface (S1 Fig and [42,43]). G208 is relatively uncoordinated based on previous structures and modeling (S1C Fig), thus CA G208R has the potential to remodel the three-fold interface of the capsid. All of this argues that the three-fold interface mutates to modulate capsid stability. CA T200I, G208R, and T216I are extremely rare in circulating HIV-1 viruses, and the residues themselves are poorly conserved compared to other CA residues (S5 Table, T200I abundance ~1% and T200 conservation rank 221/231). Thus, the inherent variability of these positions suggests they are adaptable. As we observed, if the capsid is too unstable, the three-fold interface will adapt to make the capsid more stable to compensate. We suggest that HIV-1 mutates the three-fold interface of the capsid to compensate for varying conditions in the host.

## An intact capsid lattice is required for integration into genes

Targeting integration away from transcriptionally active genes affects entry into latency and decreases reactivation from latency, characteristics that might be beneficial in persons infected with HIV-1 [11–14]. The completely unstable IP5/6-dependent HIV-1 mutants CA K170A and K203A [15] in themselves redistribute integration targeting away from transcriptionally active regions (Fig 6A and 6B). Other more stable capsid mutants (P38A, E75A, R167A, and Q219A) had no effect on ISD. Thus, the severity of capsid instability likely explains why mutants like CA K170A or K203A result in integration defects, whereas other stability mutants had no defect (Fig 6A and 6B). Underlining this point, is the ability of IPPK KO and IPMK KO to disproportionately retarget the integration of HIV-1 CA P38A and Q219A (Figs 6C and 6D and 8B, X). KO of HIV-1 integrase interacting host protein lens epithelium-derived growth factor (LEDGF/ p75) leads a similar consequence in integration targeting (S5C Fig). However, LEDGF/p75 KO additionally increases targeting near CpG islands and promoters, an aspect not observed for any of the CA mutants that we tested (S5D Fig and S4 Table, and [8,10]).

Our data demonstrate that complete capsid destabilization results in rearrangement of ISD that is unique from CPSF6 KO and the CA N74D mutation (S5C Fig). We suggest that conditions that severely destabilize the capsid lead to an absence or unstable association of CA with the PIC (Fig 8A and 8B, X). The interaction of CA with CPSF6 targets integration to transcriptionally active genes located in SPADs [8,10]. Thus, the disruption of CA association with the PIC would lead to a premature or possibly progressive loss of CA-bound CPSF6, explaining the failure of HIV-1 to effectively target genes and limited SPAD targeting for our IP5/6-dependent CA mutants in WT or IP5/6-depleted cells (Figs 6G and S5C). Stabilization of the IP5/6-dependent mutant capsids with the T200I mutation rescues ISD changes, suggesting that the presumed defects in the PIC observed for the IP5/6-dependent HIV-1 mutants were corrected. Thus, our data hint that the composition or structure of the PIC may be affected by mutations conferring IP5/6-dependence, especially in IP5/6-depleted cells (Fig 8B, X).

## BST-2 promotes the replication of HIV-1 CA mutants

HIV-1 acquired truncated forms of Vpu upon replication of CA Q219A, K170A/T200I, and T200I/K203A mutants in culture (Figs 3A and 7A–7C). Further, loss of Vpu antagonization of BST-2 with the A14L/A18L Vpu mutation promoted the replication of HIV-1 CA Q219A and T200I/Q219A (Fig 7E and 7F). Therefore, Vpu antagonization of BST-2 limits HIV-1 adaptation in culture, especially for otherwise replication deficient CA mutants. Loss of Vpu restriction of BST-2 function promotes cell-to-cell transmission of HIV-1 [60]. In a previous study,

preventing HIV-1 replication by mutating the p6 domain of Gag or antiretrovirals caused HIV-1 to acquire Vpu and/or Env mutations that promoted replication by cell-to-cell transfer [61]. Cell-to-cell transfer increases the amount of virus up taken into the target cell and can limit the effectiveness of antiretroviral drugs in cell culture [62–65]. Therefore, the enhanced cell-to-cell transfer caused by loss of Vpu restriction of BST-2 function may help the virus overcome any residual target cell infectivity defect resulting from the CA mutation(s) by increasing the amount of virus inoculated per cell during spread. It is unclear how HIV-1 spread in culture affects cellular IP5/6 levels, thus it is conceivable that during spread, the IP5/6 levels of target cells are decreased, thereby leading to restriction of mutants like HIV-1 CA P38A and Q219A. This could explain why these two mutants infected WT CEM target cells relatively well in single-cycle assays but failed to efficiently establish spreading infections (Fig 3A and [17]).

## Capsid stability influences LCV efficiency and potency against HIV-1

LCV binds to an intra-hexamer interface to block host factor interactions with CA and disrupt the capsid, thereby inhibiting infection [28,29]. Our IP5/6-dependent HIV-1 mutants had diverse responses to LCV (Fig 5). Both of the hyperstable CA T200I and E45A mutants were less sensitive to LCV than WT HIV-1. All of these mutations are distant from the LCV binding site, indicating that sites other than the LCV binding site confer resistance to LCV, a conclusion consistent with our previous studies using LCV and PF74 [29,30]. We cannot exclude the possibility that some of the CA mutations affect LCV binding. For mutation affecting the three-fold interface (T200I, K203A, and Q219A) an indirect effect of the mutation on LCV seems unlikely, since LCV binds at the NTD-CTD interface. While allosteric effects of the three-fold interface substitutions are possible, such effects have not been observed on the binding of this family of CA inhibitors, which includes PF74. By contrast, resistance to small molecule capsid inhibitors can occur without affecting their binding. For example, the E45A substitution in CA renders HIV-1 highly resistant to PF74, yet E45A does not appear to inhibit binding of the compound to the viral capsid [30]. Thus, disrupting inter/intra-hexamer interactions with new CA inhibitors should improve LCV performance in patients. We suggest that the CA interfaces modulated by P38 and Q219 represent attractive targets for next generation CA inhibitors.

## Materials and methods

### Plasmids and viruses

Mutant HIV-1 clones in this study were generated by overlap extension PCR using Q5 DNA polymerase (NEB) and the primers detailed in S6 Table. Amplified regions were ligated into pHIV-1-GFP(Δ*env*Δ*nef*) [66] or pR9 using *BssH*I/*Spe*I for N-terminal CA mutations, *Spe*I/*Apa*I for C-terminal CA mutations, or *EcoR*I/*BamH*I for Vpu mutations (S6 Table). Recombinant plasmid clones were sequenced by Sanger sequencing (GenHunter) using primers that encompass the entire insert (S6 Table). PEI (Polysciences, 23966) was used to transfect HEK293T cells as previously described [29]. HIV-1$_{GFP}$ was produced by co-transfecting pCMV-VSVg and pHIV-1-GFP into HEK293T cells. pR9 was transfected alone to make HIV-1$_{R9}$. The supernatant containing HIV-1 was clarified by passage through a 0.45 μM PES syringe filter (Sarstedt). The viruses in Figs 1 and 3A were concentrated by ultracentrifugation [24]; all other figures used unconcentrated virus. HIV-1 CA WT stocks were assayed for p24 by ELISA [24] and exogenous RTN following inactivation with 0.2% Triton X-100 [29]. Because several CA mutants affect mouse anti-CA (clone 183-H12-5C) binding to its epitope [16,36], exogenous RTN (along with a corresponding WT HIV-1 standard with a known p24

concentration) was used to assay CA mutant quantity. $^3$H incorporation by RT was measured using a MicroBeta Trilux scintillation counter (Perkin Elmer).

### Cells and infection

Culture of HEK293T, CEM, and MT-4 lines and derivation of CRISPR knockout clones was as previously described [24]. IPPK$_{Vector/Flag-IPPK}$ and IPMK$_{Vector/IPMK-Flag}$ cell lines corresponded to clone E2 and D6, respectively, for CEM cells and clone B6 and C5, respectively, for MT-4 cells [24]. For target cell infection studies using HIV-1$_{GFP}$, 4 or 8 ng p24 of HIV-1$_{GFP}$ was inoculated into cultures containing 200,000 MT-4 or CEM cells, respectively. At 48 hours post infection, cells were fixed in 2% paraformaldehyde (MP Biochemicals) and analyzed by flow cytometry on a FACS Canto II (BD) [29]. Infection in the presence of LCV (purchased from MedChemExpress) were performed as described [29] using the concentrations indicated in Fig 5. Flow cytometry data were analyzed using flowCore [67] and flowDensity [68] in R, as previously described [29]. Dose response curves were fit as previously described [29]. For all figures that used IPPK KO or IPMK KO cell lines, data were internally normalized to the HIV-1 CA WT virus to account for differences between target cell infection between the cell lines [29].

HIV-1 RTN in cells was assayed using by qPCR as described [29] using Maxima SYBR Green/ROX qPCR Master Mix (Thermo) and a Mx300P qPCR machine (Stratagene). To account for minor differences between IPMK KO$_{Vector}$ and CEM WT$_{Vector}$ cells (S3B Fig), the amount of the indicated RTN product was normalized to HIV-1 CA WT control virus.

For HIV-1 spread assays, 100,000 CEM cells were infected with the indicated amount of HIV-1$_{R9}$. In Fig 7D to 7F, cells were inoculated with 10 ng p24 of the indicated HIV-1$_{R9}$ viruses. At 1 day post infection the cells were pelleted at 800 x g for 5 minutes, supernatants were removed, and cells were resuspended in fresh medium. Every 2 to 3 days, 60% of the culture medium was collected, subjected to exogenous RTN assay to quantify accumulated virus, and replenished. Additionally, on days of media collection, the cells were split as needed. Upon detection of virus growth, a fraction of the cells was harvested for genotyping. In Fig 3A, P2-B was spin inoculated by spinning cells and virus at 2,500 rpm for the first 2 hours of infection.

### HIV-1 core purification and uncoating

HIV-1 core purification was performed as described [44]. Uncoating reactions were as described by [29] with the modification that the uncoating buffer lacked dNTPs, MgCl$_2$, and IP6. Briefly, uncoating reactions were incubated at 37˚C for 0, 1, or 2 hours at which point 50 μL of the reaction was diluted into 450 μL in ice-cold uncoating buffer and pelleted at 45,000 rpm for 30 minutes at 4˚C [29]. Pelleted cores were resuspended in 500 μL of uncoating buffer. The pellet (core-associated) and supernatant (soluble CA) fractions were assayed for p24 by ELISA. At each timepoint the % intact core was calculated by dividing the pellet associated CA by the total CA. To control for differences between the fraction of WT and T200I cores uncoated after no incubation, data were normalized by the % intact cores at each timepoint by the respective % intact cores after no incubation.

### Assay of HIV-1 uncoating using permeabilized virions

66 μL of HIV-1 particles (>66 ng p24) in complete DMEM were diluted with 154 μL of ice-cold uncoating buffer (final concentrations in reaction: 20 mM Tris pH 7.6, 0.2% Triton X-100, 150 mM NaCl, 1 mg/mL BSA, 0.5 mM DTT, and IP6 (0, 10, 100, or 500 μM)). Following 0 and 1 hour of incubation at 37˚C, 100 μL of the uncoating reaction was removed and added

to a tube containing 400 μL of ice cold uncoating buffer lacking Triton X-100. The diluted uncoating reaction was underlaid with 200 μL 20% sucrose in 20 mM Tris pH 7.6, 150 mM NaCl, and the corresponding concentration of IP6. HIV-1 cores were pelleted for 30 minutes at 4˚C at 45,000 rpm (124,740xg) using a Beckman TLA55 rotor, and the supernatant was discarded. Pelleted cores were resuspended in 500 μL ice cold uncoating buffer lacking IP6 and assayed for p24 by ELISA. For each HIV-1 variant assayed for uncoating, a reaction lacking Triton X-100 was assayed in tandem. Values shown in panel C correspond to the amount of p24 in the pellet divided by the amount of p24 in the virion.

## Endogenous RTN using permeabilized HIV-1

HIV-1 particles (~20 ng p24 per reaction) were treated with 2 μg/ mL DNAseI in complete DMEM supplemented with 10 mM MgCl2 for 1 hour at 37˚C to remove contaminating plasmid DNA. Particles were diluted to 500 μL in STE (10 mM Tris-HCl pH 7.4, 100 mM NaCl, and 1 mM EDTA) and 200 μL of 20% sucrose was gently underlaid. Virions were pelleted at 45,000 rpm (124,740xg) for 30 minutes at 4˚C using a Beckman TLA55 rotor, and the supernatant was removed and discarded. Pelleted virions were resuspended in 30 μL of 1x ERT buffer (20 mM Tris pH 7.6, 2 mM MgCl2, 150 mM NaCl, 1 mg/mL BSA, and 0.5 mM DTT) per reaction. A small aliquot of the resuspended virus solution was assayed for exogenous RTN to control for differences in input into the ERT reaction. ERT reactions were initiated by adding 20 μL of permeabilized ERT reaction buffer (1x ERT buffer containing 2.5x concentrations of Triton X-100 (0.5%), dNTPs (0 or 1 mM), and IP6 (0, 25, 250, or 1250 μM)). Reactions were incubated at 37˚C for 17 hours. DNA was extracted by silica column purification and quantified by qPCR as previously described [29]. For each virus an ERT reaction lacking dNTPs was included to determine the background of the assay. Values shown in Fig 4D are the average of 3 or more independent ERT reactions. ERT product values are normalized by exogenous RTN assay followed by subtraction of assay background using the no dNTP control.

## HIV-1 genotyping and protein alignment

Genomic DNA was extracted using the DNeasy Blood & Tissue Kit (Qiagen). 400 to 800 ng of DNA were used as input to amplify two partially overlapping fragments of the HIV-1 genome, 3,126 (before *gag* to the middle of *pol*) and 3,824 (*pol* to *env*) basepairs in length (see S6 Table for primer sequences) using NEBNext Ultra II Q5 Master Mix (New England BioLabs). The amplified HIV-1 loci were purified using the QIAquick PCR Purification Kit (Qiagen) and sequenced by Sanger sequencing (GenHunter) using primers specified in S6 Table. Mutations were identified by alignment with the HIV-1$_{R9}$ genome. Heterozygous HIV-1 loci were found and parsed into individual HIV-1 variants using the Poly Peak Parser online tool [69]. HIV-1 Vpu variants were translated and aligned using Clustal Omega [70]. All mutations were verified by aligning to HIV-1$_{R9}$ and by analyzing the Sanger sequencing chromatogram file using Chromas 2.6.6 (Technelysium Pty Ltd).

## HIV-1 integration site sequencing

Prior to inoculation, the HIV-1$_{GFP}$ viruses shown in Fig 6 were treated with DNAseI for 1 hour at 37˚C in a reaction containing 1xRPMI 1640 media (Gibco) supplemented with 10U/ mL penicillin-streptomycin (Gibco), 10% FBS, 10 mM MgCl$_2$ (Sigma) and 20 μg/ mL DNAseI (Sigma). 200,000 WT, IPPK KO, or IPMK KO CEM cells were infected at a maximal multiplicity of infection of 0.4 as measured by a FACS Canto II flow cytometry (BD) or a Countess II FL cell counter (Life Technologies). After infection, cultures were split as needed. At 4 days post infection, cells were pelleted at 800 x g for 5 minutes, washed one time with 1 mL of 1 x

PBS, and then frozen at –80˚C. DNA was extracted using the DNeasy Blood & Tissue Kit (Qiagen).

To prepare Illumina-ready libraries, 5 to 10 μg of genomic DNA was digested overnight at 37˚C with *Mse*I and *Bgl*II (New England BioLabs). To make the annealed *Mse*I half adapter, 10 μM D70X_(MseI) (where X is 1, 2, 3, 4, or 5, S6 Table) Illumina adapter oligonucleotide was annealed to 10 μM adapter complement oligo (S6 Table) in 10 mM Tris pH 8.0, 0.1 mM EDTA, and 50 mM NaCl in a Thermocycler (95C for 40 seconds followed by 75 cycles decreasing 1˚C per 55 second cycle ending at 20˚C). After purifying the digested DNA using the QIAquick PCR Purification Kit (Qiagen), 2 μg of DNA were ligated overnight at 12˚C to the 1.5 μM annealed *Mse*I half adapter in a 100 μL reaction containing 1,600 U of T4 DNA ligase (New England BioLabs) and 1 x T4 DNA ligase buffer (New England BioLabs). Enzyme and buffer were removed using the QIAquick PCR Purification Kit (Qiagen), and DNA was subjected to two rounds of semi-nested PCR using the primers detailed in S6 Table and NEBNext Ultra II Q5 Master Mix (New England BioLabs). Between rounds of PCR, DNA was purified using SPRI DNA purification magnetic beads (Applied Biological Materials Inc., 1.4 beads: 1 DNA ratio). After the second round of PCR, DNA was again purified using the QIAquick PCR Purification Kit (Qiagen). The resulting library was subjected to 150 bp paired end sequencing on a NovaSeq 6000 at a minimum of 3,000,000 reads by the VANTAGE core at Vanderbilt University Medical Center.

FASTQ files were processed as follows to find unique integration sites: Residual Illumina adapter TruSeq sequences were removed from the FASTQ files using BBDuk [71]. Sequences containing the terminal 15 nucleotides of the HIV-1 U5 3' LTR sequence were kept and trimmed, removing the LTR sequence using custom python scripts. Sequence pairs were mapped to human genome build hg38 and HIV-1$_{NL4-3}$ using HISAT2 [72]. Custom python scripts were used to retain reads that uniquely map solely to the human genome at a mapping quality of >40 and make positional duplicates unique, thus outputting a file containing a list of unique genomic coordinates of HIV-1 integration sites. For each Illumina sequencing run, the list of positionally unique integration sites was compared across libraries using BedTools [73]. Integration sites occurring in multiple libraries were removed, creating a list that was annotated using BedTools [73]. Genomic locations of CpG island, transcription start sites, and NCBI ReqSeq genes were downloaded from the UCSC genome browser. The locations of LADs and SPADs were as described by reference [9] and [10], respectively. The *in silico* generated Random integration control (RIC) was produced as described previously [74]. HIV-1 integration sites for WT, LEDGF/p75 KO, CPSF6 KO, and LEDGF/CPSF6 double KO HEK293T cells were from [8] and were mapped and annotated as described above. Lists of the positionally de-duplicated human genomic integration sites are included in S1 Data through S6 Data.

## Calculation of CA amino acid frequency

Gag protein (complete sequence) alignments for HIV-1/SIV$_{cpz}$ were downloaded from the Los Alamos National Laboratory HIV-1 sequence database and comprised of 10,591 Gag sequences (https://www.hiv.lanl.gov/content/sequence/NEWALIGN/align.html). The output FASTA file was processed using a custom python script that positionally tabulates the number of appearances of each amino acid. The output file was then imported into Excel (Microsoft) to calculate the positional amino acid frequency and calculate the amino acid frequency rank (S1 Table).

## Inter-CA bond identification and structural mapping

In Fig 1E, the X-ray structure of the native CA hexamer (PDB 4XFX, [43]) was used to map residues tested for IP5/6 dependence using UCSF Chimera [75]. The HIV-1 trimer interface in

Fig 1F was modeled in UCSF Chimera by aligning the NMR structure of the full-length HIV-1 CA protein (PDB 6WAP, [76]) to a portion of the cryo-electron microscopy structure of the HIV-1 capsid (PDB 5MDG, [77]). Figs 2A, 2B, S1A and S1B show an atomic model of the HIV-1 capsid (PDB 6J3Y, [42]) with the indicated residues mapped onto it using UCSF Chimera. For inter-CA bond identification, the find contacts command in UCSF Chimera was used with default parameters on an all atoms models of the HIV-1 capsid (PDB 6J3Y, [42]).

## Statistics and data availability

Excluding HIV-1 spread assays which were performed two times, all experiments were performed independently at least 3 times with data points showing the average and error bars denoting standard deviation. An ANOVA test was performed prior to all subsequent statistical analyses. In Figs 1, 2, 3, 4, 5, 6C and 6D a one-tailed T test assuming unequal variances was used to compare conditions. All multiple comparisons were corrected for using the Hommel method in R. In Fig 6A and 6B, Dunnett's test was performed in R to compare the HIV-1 CA WT control to all mutant conditions, thereby correcting for multiple comparisons. p value thresholds: * $p < 0.05$, ** $p < 0.01$, *** $p < 0.001$, and **** $p < 0.0001$.

All relevant data pertaining to the figures has been depositing into the Dryad database (see reference [78] for citation and doi (https://doi.org/10.5061/dryad.5mkkwh79k))).

## Supporting information

**S1 Fig. Inter-CA interactions of selected CA residues.** (A and B) Enlarged versions of the right panel of Fig 1A and 1B that portrays the inner (A) and outer (B) surface of an atomic model of the HIV-1 capsid (PDB 3J3Y). (C) Intermolecular CA interactions of IP6-dependent and suppressor residues using an atomic resolution model of the HIV-1 capsid (PDB 3J3Y). The y-axis displays CA residues identified by UCSF Chimera predicted to form intermolecular CA contacts. Bubble size is indicative of frequency of the intermolecular interaction in the model.
(TIF)

**S2 Fig. HIV-1$_{R9}$ CA Q219A selects for CA T200I and Vpu W22stop during replication in cells.** (A—E) Sanger sequencing chromatograms at the indicated HIV-1 loci during passage 1 (P1) and subsequent passages (P2 and P3) of the HIV-1$_{R9}$ CA Q219 (25 ng p24). P2-A and P2-B are derived from the same P1 HIV-1$_{R9}$ CA Q219A virus shown in Fig 3A. P3-A and P3-B originate from virus released from P2-A and P2-B, respectively.
(TIF)

**S3 Fig. HIV-1 CA T200I does not rescue HIV-1 CA Q63A/Q67A.** (A) Target cell infection of WT$_{Vector}$, IPMK KO$_{Vector}$, or IPMK KO$_{IPMK-Flag}$ CEM cells by flow cytometry of the indicated HIV-1$_{GFP}$ CA mutants. (B) RTN at 8 h by qPCR in the indicated CEM cell lines.
(TIF)

**S4 Fig. Bar graphs of LCV dose response shown in Fig 4.** (A) LCV inhibition of the indicated HIV-1$_{GFP}$ CA mutant infection of WT CEM cells. Black asterisk(s) show statistical significance relative to HIV-1$_{GFP}$ WT. Colored asterisk(s) denote significance of the indicated single mutant compared to the respective colored double mutant. Values shown are the average of at least 3 independent experiments. Significance levels: $p < 0.05$ *, $p < 0.01$ **, $p < 0.001$ ***, and $p < 0.0001$ ****.
(TIF)

**S5 Fig. Comparison of HIV-1 ISD in HEK293T and CEM cell lines.** (A-D) HIV-1 ISD results represented by various genomic annotations. Similar to Fig 5 panels E to G, except HEK293T WT and KO cell lines are included for comparison. Abbreviations: LEDGF/p75 KO (LKO), CPSF6 KO (CKO), and LEDGF/p75 and CPSF6 KO (DKO).
(TIF)

**S1 Table. Effect of IP5/6-dependent HIV-1 CA mutants on LCV potency.** Lenacapavir dose response parameters for the indicated HIV-1 CA mutants.
(XLSX)

**S2 Table. Summaries for each integration site sequencing library.** Number of unique integrations for each library.
(XLSX)

**S3 Table. Summary statistics for HIV-1 CA mutant integration into WT CEM cells.** Average percentage of integrations (with standard deviation) into the indicated genomic annotations.
(XLSX)

**S4 Table. Summary statistics for HIV-1 integration into various WT and KO cell lines.** Average percentage of integrations into the indicated genomic annotations.
(XLSX)

**S5 Table. CA amino acid frequencies of selected CA residues.** Amino acid frequency table for the indicated residues of HIV-1 CA. Sequences are from the Los Alamos National Laboratory HIV-1 sequence database.
(XLSX)

**S6 Table. Oligonucleotides used in this study.** Table of oligonucleotide sequences along with their use in this study.
(XLSX)

**S1 Data. Genomic coordinates (hg38) in bed file format of unique integrations into the genome of WT CEM cells (package A).**
(TAR.GZ)

**S2 Data. Genomic coordinates (hg38) in bed file format of unique integrations into the genome of WT CEM cells (package B).**
(TAR.GZ)

**S3 Data. Genomic coordinates (hg38) in bed file format of unique integrations into the genome of IPPK KO CEM cells.**
(TAR.GZ)

**S4 Data. Genomic coordinates (hg38) in bed file format of unique integrations into the genome of IPMK KO CEM cells.**
(TAR.GZ)

**S5 Data. Genomic coordinates (hg38) in bed file format of unique integrations for the RIC.**
(TAR.GZ)

**S6 Data. Genomic coordinates (hg38) in bed file format of unique integrations into the genome of WT, LKO, CKO, and DKO HEK293T cells.**
(TAR.GZ)

## Acknowledgments

The following reagent was obtained through the NIH HIV Reagent Program, NIAID, NIH: Efavirenz, HRP-4624, contributed by DAIDS/NIAID.

## Author Contributions

**Conceptualization:** Gregory A. Sowd.

**Data curation:** Gregory A. Sowd.

**Formal analysis:** Gregory A. Sowd.

**Funding acquisition:** Gregory A. Sowd, Christopher Aiken.

**Investigation:** Gregory A. Sowd, Jiong Shi, Ashley Fulmer.

**Methodology:** Gregory A. Sowd.

**Project administration:** Gregory A. Sowd.

**Resources:** Gregory A. Sowd, Christopher Aiken.

**Software:** Gregory A. Sowd.

**Supervision:** Gregory A. Sowd, Christopher Aiken.

**Validation:** Gregory A. Sowd.

**Visualization:** Gregory A. Sowd.

**Writing – original draft:** Gregory A. Sowd.

**Writing – review & editing:** Gregory A. Sowd, Ashley Fulmer, Christopher Aiken.

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
