## [Decision Letter · Decision Letter 0]

10 Mar 2023

Dear Dr Sowd,

Thank you very much for submitting your manuscript "HIV-1 capsid stability enables inositol phosphate-independent infection of target cells and promotes integration into genes" for consideration at PLOS Pathogens. As with all papers reviewed by the journal, your manuscript was reviewed by members of the editorial board and by several independent reviewers. In light of the reviews (below this email), we would like to invite the resubmission of a significantly-revised version that takes into account the reviewers' comments.

We have received reviews from 4 referees, and all of them are supportive of the paper. However, there are important issues that need to be resolved in a revised manuscript.

We cannot make any decision about publication until we have seen the revised manuscript and your response to the reviewers' comments. Your revised manuscript is also likely to be sent to reviewers for further evaluation.

Sincerely,

Owen Pornillos

Guest Editor

PLOS Pathogens

Susan Ross

Section Editor

PLOS Pathogens

Kasturi Haldar

Editor-in-Chief

PLOS Pathogens

orcid.org/0000-0001-5065-158X

Michael Malim

Editor-in-Chief

PLOS Pathogens

orcid.org/0000-0002-7699-2064

We have received reviews from 4 referees, and all of them are supportive of the paper. However, there are important issues that need to be resolved in a revised manuscript.

Reviewer's Responses to Questions

**Part I - Summary**

Reviewer #1: This manuscript describes a characterisation of mutations in the HIV capsid that confer dependence on IP5/IP6 in the target cell, and compensatory mutations that arise during their adaptation to spreading infection. A mechanistic explanation is offered for the various phenotypes, and the importance is contextualised by the recent development of lenacapavir – a drug which targets the capsid.

A strength of this manuscript is the molecular virology. The identification of CA mutants P38A, K170A, K203A, and Q219A as being IP5/IP6-dependent is compelling, the experiments identifying of T200I and its effects are well-executed, as are the integration targeting measurements.

While the phenomenology is interesting, a weakness of the manuscript is the mechanistic insight, particularly with respect to capsid stability arguments and the role of Vpu. The only stability measurement presented is in Figure 4 (sucrose gradient purification of CA cores), and it is not compelling (see below). The role of BST-1 and cell-to-cell transmission requires more exploration. Furthermore, the arguments are, at times, difficult to follow: sometimes there appear to be missing mutants, or the rationale for their omission is not provided; wording can be convoluted requiring multiple reads; and figures often don’t encode the key information in a digestible way relying heavily on descriptive text (legends and main text) for their interpretation. The weakness of the mechanistic insight is highlighted by the fact that the data ‘hint’ at interpretations multiple (3) times in the text.

Reviewer #2: Sowd et al. demonstrate that unstable capsid mutants of HIV-1 have reduced infection, RTN, integration and replication in cells where inositol phosphate kinases IPMK and IPPK are knocked-out. This is a hugely significant finding as it suggests that IP5 and IP6 remain important for HIV-1 infection post-assembly and prior to integration in target cells. The data is of excellent quality and the experiments well executed. A particularly noteworthy result is the finding that the IP5/IP6 sensitivity of unstable capsid mutants can be compensated for by a second site suppressor, T200I. Overall, this is an important study and will be of considerable interest to the field. I am therefore highly supportive of publication. A few minor suggestions are listed below but these are for consideration by the authors only and not conditional for acceptance.

Reviewer #3: Sowd and colleagues investigated a select group of CA mutants that adversely affect core stability for their ability to infect IP5/IP6 depleted target cells. Interestingly, they identified four substitutions (P38A, K170A, K203A and Q219A) in CA, which exhibited >100-fold reduced infectivity in IP5/6 depleted cells vs control cells. Furthermore, the authors serially passaged HIV-1(Q219A CA), which resulted in emergence of T200I CA substitution as well as Vpu truncation. These are thought-provoking findings. However, I am failing to find sufficient evidence for their main claim that the T200I CA change enhances capsid stability and thereby counteracts destabilizing effects of Q219A CA (see below).

Reviewer #4: The manuscript by Sowd et. al., is well written and presents an impressive amount of data. The study of how mutations in HIV CA in combination with depletion of IP6 and IP5 from cells, and/or the addition of capsid targeting compounds alter productive infection of physiologically relevant T-cells is both timely and important. The studies novelty residues in the effect of destabilizing mutations in the context of depletion of the critical stabilizing small molecules IP5/6. Mutations in CA that make HIV more reliant on target cell IP5/6 used to interrogate the role of IP5/IP6 on infection, and to identify compensatory mutations. Overall, this manuscript presents a large amount of data that is reasonably interpreted. However, the conclusions are too strongly stated, and the background data related to IP5/IP6 depletion are too often written as though depletion is equal to elimination.

**Part II – Major Issues: Key Experiments Required for Acceptance**

Reviewer #1: The most significant issue is Figure 4A. As it stands based on the colour scheme, the T200I mutant does not form stable cores. It is possible this is an error (as two of the colour schemes in the key do not appear in the graph), but if it is not, this undermines the entire mechanistic argument that T200I provides a stabilisation effect to the capsid. It would be preferable to see some more compelling evidence regarding lattice stability. Electron micrographs of the mutants to support whether they have any morphological defects would be useful. In vitro CA assembly studies with IP6 and LCV to show that their relative assembly competencies and responsiveness to these compounds would also be helpful. Single particle uncoating assays would be ideal (accepting that this is a specialised imaging technique), but some method to quantify relative capsid stabilities is important given that ‘stability’ features so prominently in the title and the proposed model.

The Vpu truncation experiments point to a possible role of BST-2 in promoting Q219A replication. This interpretation should be tested by observing the effect of BST-2 overexpression/depletion.

Figure 8 needs to be completely remade. A flow of information that starts in the middle, immediately diverges, and mirrors itself with multiple pathways that get further away from each other, but need to be compared, is a poor way to compare and contrast scenarios. The details of superscripts (T200I label reverses the steps?), dashed lines, colours, are too confusing.

Reviewer #2: There are no major issues.

Reviewer #3: 1. The results related to relative stability HIV cores are presented in Fig 4. The graphs in Fig 4A appear to be mislabeled as they do not correspond to the description of these results (see lines 246-249). Based on the description, the solid green line would be T200I rather than Q219A. Also, I cannot tell which mutants are shown by dashed and solid blue lines, but the description indicates that the cores with Q219A and Q219A/T200I mutations could not be purified. Even after the authors correct their careless mistakes, how can one conclude from these results that Q219A/T200I is more stable than Q219A?

2. To somehow justify their claim about potentially stabilizing effects of the T200I change, they compare WT vs T200I CA cores in Fig 4B. These assays compare WT with T200I CA rather than Q219A vs Q219A/T200I. Furthermore, I doubt that this assay is quantitative enough or the observed differences between WT vs T200I CA cores are substantial enough to justify their claim. Did they do control experiments in the presence of IP6 to show that both WT and T200I cores remain similarly stable during 2 hrs? I would suggest using more sophisticated assays (for example see PMID: 27322072) to quantitate effects of the T200I CA change. Once again, what matters is in the difference between Q219A vs Q219A/T200I.

3. The authors talk exclusively about effects of these mutations on core stability. Did they measure how these substitutions affect virion maturation? I would like to see TEM images of Q219A vs Q219A/T200I as well as WT vs T200I virions. The quantitative comparison of these virions may shed light whether the examined substitutions affect late steps of HIV-1 replication/formation of properly mature, conical cores.

4. Their results with lenacapavir are overinterpreted by solely looking at them through a prism of the capsid stability. Did they check how these CA substitutions affect lenacapavir binding to CA hexamers or monomers? These amino acids are not positioned in the inhibitor binding pocket but indirect effects of various substitutions on the inhibitor binding must be given an important consideration. I am not asking to do these additional biochemical experiments, but the authors should markedly tone done their speculation and acknowledge other possibilities. For example, it is well documented that lenacapavir binds pre-formed, stabilized hexamers with substantially higher affinity than CA monomers (PMID: PMID: 32612233). These findings are not considered in the discussion.

Reviewer #4: 1) The description of the data and the conclusions are reasonable. However, the figure presentation is difficult to interpret. A solution would be to provide a key of the mutations mapped onto the structure or a depiction of the lattice. The authors attempt this in one figure, but it is too small, and not nearly informative enough. The addition of such mutation maps could drastically improve the readers understanding of the interpretation of the results, and the rational for certain experiments.

1) The manuscript correctly reports that IP5/IP6 depletion in cells that are being infected does not significantly alter the outcome. If this is true, what is the purpose of enriching IP5/IP6 in the immature virus particle? If IP5/IP6 is needed to purify capsid in vitro, why is it not needed in receiving cells? Given that the KO's used do not eliminate both IP5 and IP6, is it possible that the remaining levels are sufficient for capsid? If so, how can WT be not dependent, but the mutants used be dependent on IP6.

A simple solution would be to soften some of the language used throughout the manuscript related to this.

Both mutations R18A and K25A are known to prevent or alter assembly of capsid, and subsequent infectivity. How do the stabilizing mutations identified at the three-fold interface respond in the context of R18A? What are the implications of these two mutations in the current study?

**Part III – Minor Issues: Editorial and Data Presentation Modifications**

Reviewer #1: Line 160 – WT is not presented in the figure

Line 181 – Why was Q219A chosen for spreading infection experiments, and not the other mutations?

Line 216 – ‘a subset’ of mutants was assayed. Why was P38A omitted from Figure 3D? It is present in all other panels in Figure 3. Also, infection is plotted on a log axis, but reverse transcription on a linear axis (Figures 1 and 3). Given that some of the infectivity defects occur over 5 logs, similar defect levels in RT are difficult to discern on a log axis.

Line 274 – ‘…K170A mutant, which contains a highly unstable capsid,…’. This statement is not substantiated by any relative stability measurement.

Line 291-294 – Rewrite this sentence, it is difficult to follow.

Line 315 – ‘Nonetheless, HIV-1 pre-integration complexes (PICs) targeted SPADs at much greater frequency than CA N74D (Figure 6B)’. This sentence is unclear in this context.

Line 332 – 334 – This sentence required multiple readings, remains unclear and the following conclusion is uncompelling. It is difficult to see how T200I counteracts defects in integration targeting when it increases SPAD integration for P38A and decreases it for Q219A in the IPMK KO cells. And again, why were two of the mutants (K170A and K203A) omitted from this experiment?

Reviewer #2: 1. Figure 1. The legend is confusing; (F) refers to quantification of early RTN produces but this is labelled (D) in the figure. Likewise, (D) and (E) refer to positions of altered residues mapped onto the hexamer or three-fold lattice but in the figure this is (E) and (F). Perhaps (D) and (F) are mixed up?

2. Was the passage (Figure 3A) of Q219A done in WT CEM cells? If so, why was T200I selected given that it does not increase infection by Q219A (Figure 3B)?

3. Line 230-231: As the FLM RTN signal is so weak for K203A and K203A/T200I, I think it should also be considered that any difference will be difficult to determine. Therefore it doesn't necessarily mean that T200I must be enhancing a step post-RTN. This data alone isn’t really enough to support the assertion that IP5/IP6 likely affects other stages of target cell infection, as stated in Lines 234-235.

4. Figure 3D. Given that other experiments were done with P38A, it’s a pity data isn’t shown for this mutant here. As P38A is arguably one of the more infectious mutants, its inclusion here would be of interest.

5. Figure 4A. I think the legend is incorrect; presumably the green circle and solid line is T200I? The blue circle solid and dotted lines are presumably Q219A and Q219A/T200I respectively?

6. Figure 4B. The uncoating experiment was carried out in the absence of IP6 but it would be useful to know whether there is any difference in the presence of IP6, maybe at longer time points?

7. Lines 303-304. “A spectrum of ISD severities…based on which aspect(s) of CA biology are affected by the mutant”. It is unclear what specific data is being used to draw this conclusion. Please clarify which aspect of CA biology each mutant is affecting.

8. Lines 313-316. Please explain the significance of K203A showing a greater %gene defect than N74D but a smaller SPAD defect and what would cause this?

9. Lines 327-332. What is the basis for describing the integration of IP5/6-dependent mutants in IP5/6-depleted cells as substantially affecting %genes but only modestly affecting SPAD? The data in Fig6C&D suggests both are equally significantly affected?

10. Figure 6. A conclusion from these experiments is that “upon IP5/6 depletion, IP5/6-dependent mutants have a unique ISD” but is it not the case that they have a similar ISD to unstable mutants? For instance, K170A shows a similar reduction in % gene integration as P38A in IPMK KO cells.

11. Related to the previous point, it would be important to test whether K170A shows any further reduction in % gene integration in IPMK KO cells, particularly as this mutant is associated with a reduction in infectivity (Figure 2B).

12. An unstable capsid in IP5/6-depleted cells would logically be expected to show defects downstream of RTN because it may collapse after RTN but before integration. Therefore in order to conclude that there are specific integration site defects, rather than merely a linear correlation between stability and the number of proviruses, the proviral DNA levels for mutants in WT, IPPK KO and IPMK KO conditions should be shown (and similar linear correlations tested as in Figure 6E-G).

13. In general the section dealing with Figure 6 could be improved by explaining why the phenotypes of certain mutants suggest they impact different aspects of CA biology. I assume the purpose here is to differentiate mutants that impact nuclear import vs integration but it is not clearly spelled out.

14. Figure 7. A further CA mutation, G208R, was obtained upon passage of T200I/K203A virus. In order to determine if this mutation has actually impacted T200I/K203A fitness it would be important to test whether it alters infectivity of T200I/K203A in WT and IP5/6-depleted cells.

Reviewer #3: They state that lenacapavir is in the phase III clinical trial. The drug has already been approved by FDA for clinical use.

Reviewer #4: The authors have not cited multiple manuscripts that detail important aspects of IP5/IP6 interactions with capsid, and the effect of IP5/IP6 cellular depletion on infectivity that are highly relevant to the current study. Please correct this oversight.

Lenacapavir is now EU and FDA approved, and goes by multiple names. All names should be indicated early, and then whatever the authors chose to call the compound beyond that is fine.

In the discussion of Lenacapavir (last paragraph of the manuscript), Fig. 4 is referenced instead of Fig. 5. Lastly, the size of panels in Fig. 6E-F can be increased to increase legibility. Please double check all figure labeling and in text reference to figures.

What was the rational for using Q219A for selections as opposed to the other mutants?

What are the implications of the mutants in this study if they are mapped onto the Pentamer or the Pentamer-Hexamer 3-fold?

Typos, poor grammar, or difficult to understand:

Lines: 58, 291-294, 348, 462

424--...IP6 may readily dissociate

438--residues published, residues reported

line 462

PLOS authors have the option to publish the peer review history of their article (what does this mean?). If published, this will include your full peer review and any attached files.

Reviewer #1: No

Reviewer #2: **Yes: **Leo C James

Reviewer #3: No

Reviewer #4: No
---

## [Decision Letter · Decision Letter 1]

14 May 2023

Dear Dr Sowd,

We are pleased to inform you that your manuscript 'HIV-1 capsid stability enables inositol phosphate-independent infection of target cells and promotes integration into genes' has been provisionally accepted for publication in PLOS Pathogens.

Best regards,

Owen Pornillos

Guest Editor

PLOS Pathogens

Susan Ross

Section Editor

PLOS Pathogens

Kasturi Haldar

Editor-in-Chief

PLOS Pathogens

orcid.org/0000-0001-5065-158X

Michael Malim

Editor-in-Chief

PLOS Pathogens

orcid.org/0000-0002-7699-2064

The original reviewers have examined the revision and are now supportive of publication. Reviewer 1 suggests additional corrections that can be done at the proofing stage. Reviewer 4 suggests that the authors should also include the pentamer in interpretation. However, I do not believe this is necessary, and including the pentamer in this analysis will require further revision. To wit, the authors include the following response to Reviewer 4's comment on the original submission (What are the implications of the mutants in this study if they are mapped onto the Pentamer or the Pentamer-Hexamer 3-fold?): "This is an interesting question, as the effects of CA mutations on hexamer-pentamer interactions is an unexplored area. A pentamer model (PDB 7URN) recently published by the Pornillo’s lab (PMID 36759579) is shown below. The model uses PDB 4XFX initial model for fitting. The model does not affect our conclusions because the purpose of the residue contacts remains the same (e.g. P38 and K170 form intra-hexamer interactions. K203 and T200 form inter-hexamer interactions. Q219A is at the intra- and inter-hexamer interface). Further, we have not included this image as the initial structure (PDB 4XFX) used to fit the cryo-EM data does not capture relevant known contact information at the 3-fold interface (making the 3-fold interfaces in the model less relevant)." There are false or misleading assertions in this response: 1) The "purpose of the residue contacts remains the same (e.g. P38 ...": P38 interactions in context of the hexamer and pentamer are completely different, as discussed in detail in Schirra et al. 2) "does not capture relevant known contact information at the 3-fold interface (making the 3-fold interfaces in the model less relevant)" - again incorrect. The Schirra et al. structure shows the pentamer-hexamer/hexamer 3-fold in the capsid. I believe this perfunctory response arises from a fundamental misunderstanding of the process of modeling PDB coordinates into high-resolution cryoEM maps. Including the pentamer in the discussion of this manuscript will likely require more rounds of revision; the ms is still a substantive contribution with addressing the pentamer.

Reviewer Comments (if any, and for reference):

Reviewer's Responses to Questions

**Part I - Summary**

Reviewer #1: The authors have addressed my concerns and the manuscript is stronger, especially with respect to stability arguments.

Reviewer #3: My concerns have been addressed.

Reviewer #4: The authors have satisfactorily responded to all major and minor comments. Their attention to detail is appreciated.

One final note for the authors consideration:

When interpreting your results in the context of particle stability, I would not focus only on where these mutations map on the CA-Hexamer. Given that IP5/IP6 is also critical to pentamer formation, it is reasonable to consider your results in the context of the pentamer.

**Part II – Major Issues: Key Experiments Required for Acceptance**

Reviewer #1: None.

Reviewer #3: None

Reviewer #4: none

**Part III – Minor Issues: Editorial and Data Presentation Modifications**

Reviewer #1: Figure 1 E/F: '2 < fold change > 3' should read '2 < fold change < 3'

Figure 8: Please ensure that the dot representing IP6 is clear in the final manuscript. It is difficult to resolve in the pdf version supplied. Suggest giving it a distinctive color.

Reviewer #3: None

Reviewer #4: none

PLOS authors have the option to publish the peer review history of their article (what does this mean?). If published, this will include your full peer review and any attached files.

Reviewer #1: No

Reviewer #3: No

Reviewer #4: No

---

## [Editor Report · Acceptance letter]

30 May 2023

Dear Dr Sowd,

We are delighted to inform you that your manuscript, "HIV-1 capsid stability enables inositol phosphate-independent infection of target cells and promotes integration into genes," has been formally accepted for publication in PLOS Pathogens.

Best regards,

Kasturi Haldar

Editor-in-Chief

PLOS Pathogens

orcid.org/0000-0001-5065-158X

Michael Malim

Editor-in-Chief

PLOS Pathogens

orcid.org/0000-0002-7699-2064